# Genome-wide errant targeting by Hairy

**Kurtulus Kok[1], Ahmet Ay[2], Li M Li[3], David N Arnosti[1,4]\***

[1]Genetics Program, Michigan State University, East Lansing, United States; [2]Departments of Biology and Mathematics, Colgate University, Hamilton, United States; [3]Janelia Research Campus, Howard Hughes Medical Institute, Ashburn, United States; [4]Department of Biochemistry and Molecular Biology, Michigan State University, East Lansing, United States

**Abstract** Metazoan transcriptional repressors regulate chromatin through diverse histone modifications. Contributions of individual factors to the chromatin landscape in development is difficult to establish, as global surveys reflect multiple changes in regulators. Therefore, we studied the conserved Hairy/Enhancer of Split family repressor Hairy, analyzing histone marks and gene expression in *Drosophila* embryos. This long-range repressor mediates histone acetylation and methylation in large blocks, with highly context-specific effects on target genes. Most strikingly, Hairy exhibits biochemical activity on many loci that are uncoupled to changes in gene expression. Rather than representing inert binding sites, as suggested for many eukaryotic factors, many regions are targeted errantly by Hairy to modify the chromatin landscape. Our findings emphasize that identification of active *cis*-regulatory elements must extend beyond the survey of prototypical chromatin marks. We speculate that this errant activity may provide a path for creation of new regulatory elements, facilitating the evolution of novel transcriptional circuits.

## Introduction

Metazoan transcriptional circuitry features activation and repression signals that constitute robust regulatory networks important for the unfolding of developmental programs. In the *Drosophila* embryo, localized transcriptional repressors provide essential patterning information that establishes the primary anterior-posterior and dorsal–ventral axes of the organism. The action of transcriptional repressors is heterogeneous and can exhibit context effects; one of the most striking aspects involves the different classes of repressors that mediate distinct chromatin changes on target genes. Short-range acting proteins Snail and Knirps interfere with transcription only when their cognate binding sites are located within close range of the activator binding sites (*Gray and Levine, 1996*). These proteins interact with evolutionarily conserved corepressors that possess chromatin modifying activities (*Nibu et al., 1998*; *Payankaulam and Arnosti, 2009*). Paradoxically, these same cofactors are also recruited by another class of repressors, the long-range transcriptional repressors, exemplified by the Hairy factor (*Paroush et al., 1994*; *Barolo and Levine, 1997*; *Poortinga et al., 1998*). This protein is a founding member of the Hairy/Enhancer of Split (HES) transcription factors, which play essential roles in animal development, including segmental gene patterning in the early embryo and specification of neuronal differentiation in response to Notch signaling (*Kageyama et al., 2007*). Thus, elucidation of molecular mechanisms of Hairy activity will shed light on a number of important gene circuits that are prominently represented in key developmental pathways. The biochemical function of Hairy is associated with long-range chromatin modifications, which endow this factor with the ability to interfere with multiple *cis*-regulatory regions, including activators bound over 1 kb distal to the Hairy binding sites. The long-range effect has been proposed to be due to the recruitment of the corepressor Groucho (Gro), that can oligomerize to spread over large areas of the genome, and colocalization of HDAC to the target genes resulting in deacetylation of specific lysine residues in histones H3

**\*For correspondence:** arnosti@msu.edu

**Competing interests:** The authors declare that no competing interests exist.

**eLife digest** The genes encoded in DNA contain the instructions to make proteins and other molecules important for cell behavior. Only a fraction of genes are 'expressed' at any particular time; proteins called transcriptional repressors keep many in a silent state. One such repressor in the fruit fly is called Hairy, and its activity is essential for embryos to develop correctly. Similar Hairy-related proteins are crucial regulators of development in mammals.

A central mechanism of controlling gene expression involves the wrapping of DNA around histone proteins to form a structure called chromatin. Attaching chemical tags to histones changes how accessible the genes are within the chromatin—the more accessible the genes are, the more likely they are to be active. Some tags promote gene activation, while other tags block expression. Previous research showed that Hairy reduces gene expression by influencing which tags are added to, or removed from, the chromatin.

Kok et al. have now tracked the effects of the Hairy protein on the entire genome of *Drosophila* fruit fly embryos. This revealed the genes that Hairy directly targets and the corresponding effects this targeting has on chromatin structure. Hairy altered chromatin chemical tags over large blocks of DNA on silenced genes, with some of the changes being specific to particular genes. However, many areas of chromatin activity were not associated with changes in gene expression. Instead, many genes ignore Hairy-mediated changes in their vicinity, while in other cases chromatin changes occurred on genes that were already silent.

Previous studies have suggested that regulatory factors like Hairy frequently bind to many sites on the genome and have no function. Kok et al. now suggest that these sites—previously regarded as representing 'inert' sites—are biochemically very active. Genomic studies that label regulatory sites solely by changes to their chromatin modifications may be fooled by the apparent activity of such 'errantly targeted' sites, assuming that they are critical for gene regulation. At the same time, these sites may represent regions that are particularly likely to evolve regulatory properties. Kok et al. therefore propose that errant targeting by Hairy may help new regulatory elements to evolve that could eventually influence how genes are expressed.

and H4 (*Courey and Jia, 2001*; *Martinez and Arnosti, 2008*). In our previous studies, we showed that Hairy induced extensive tracts of deacetylation on *ftz*, a segmental patterning gene expressed early in embryogenesis (*Li and Arnosti, 2011*).

While potent in repression potential, Hairy and other long-range repressors are apparently restricted in their ability to exercise transcriptional effects by the local *cis*-regulatory context in which binding sites are located. Hairy was demonstrated to lack long-range effects on a distal RACE enhancer in the embryonic dorsal ectoderm, when Hairy binding motifs were situated in an element with activators that are restricted to mesoderm/neurectoderm regions. Furthermore, the Dorsal protein, when itself acting as a long-range repressor, is dependent on neighboring Cut and Dri transcription factor motifs to function, indicating that long-range repression complexes may require specific *cis*-regulatory grammar (*Cai et al., 1996*; *Nibu et al., 2001*).

The action of eukaryotic transcriptional repressors involves a number of biochemical activities, including direct antagonism of transcriptional activators and assembly of chromatin-associated factors that are correlated with gene silencing (*Perissi et al., 2010*). Specific types of covalent histone modifications, such as H3 and H4 deacetylation, H3K9 trimethylation and H3K27 trimethylation are correlated with repressed genes, but there is still no general understanding of how important in a quantitative sense such modifications are for inhibition of transcription at specific genes. Context effects for a particular transcriptional repressor can influence what sort and how much of a response will be generated. At a genome-wide level, specific chromatin features correlate with transcriptionally repressed genes (e.g., H3K9 and 27 methylation, reduced levels of H3 and H4 acetylation, binding of HP1), however these marks are also found within highly active loci (*modENCODE Consortium et al., 2010*). The epigenetic signature of transcriptional repression is thus context-dependent, consistent with a revised picture of the simple 'histone code' hypothesis. In the context of specific transcriptional repressors, we know little about how the context of distinct factors present at *cis*-regulatory elements shapes their action. Genome-wide information obtained from chromatin immunoprecipitation

experiments should provide information about molecular targets and action of transcription factors, however, in addition to *bona fide* regulatory targets, metazoan transcription factors typically associate with a large number of in vivo binding sites of unknown significance. Recent studies have suggested that these interactions represent off-target genomic interactions, driven by low binding specificity of transcription factors and a general affinity for open chromatin of active enhancers (*MacArthur et al., 2009*). A survey of possible 'off target' binding elements suggested that these tend to be of lower affinity and are transcriptionally inert (*Fisher et al., 2012*). As noted above, previous studies of Hairy suggested that the protein is unable to mediate transcriptional repression in the absence of other factors co-occupying regulatory elements (*Nibu et al., 2001*).

Identification of functional properties of Hairy transcends the simple biochemical elucidation of repression; this protein is representative of the regulatory factors comprising conserved gene regulatory networks (GRN) that constitute the basis of animal development. Molecular studies have demonstrated that the acquisition or loss of binding sites or entire regulatory modules appears to drive significant changes in gene expression that initiate critical evolutionary transitions, such as elaboration of novel limb structures (*Khila et al., 2009*; *Pavlopoulos et al., 2009*; *Tanaka et al., 2011*). Significantly, although relatively subtle changes have been linked to such important evolutionary innovations, it appears that functional conservation of gene expression is also compatible with major changes in the structure of transcription control regions (*Hare et al., 2008*). The constraints for reorganization of existing *cis*-regulatory elements, or appearance of such elements de novo, are poorly understood; in some cases, the exact placement of multiple transcription factor motifs is essential for transcriptional function, while the composition of other genetic switches appears to be very loosely organized (*Arnosti and Kulkarni, 2005*). The existence of a large fraction of 'off-target' binding sites both complicates the analysis of important functional links, and the interpretation of potential evolutionary changes. Thus, elucidation of the functional targets and chromatin effects of Hairy can provide important insights on the basic substance of evolutionary variation. In this study, we use genetic tools to mediate induction of Hairy on a short time scale, permitting us to identify direct regulatory targets and chromatin effects on a genome-wide level. In addition to identifying common features of Hairy repression mechanisms across many targets, we also show that this protein exerts pervasive biochemical activity to change chromatin states at many loci unlinked to gene expression, revealing a possible pathway to evolution of novel gene regulatory connections.

## Results

### Genome-wide transcriptional regulation by Hairy

To study transcriptional repression at the genome-wide level at this important developmental stage, we profiled changes in transcriptome, epigenome and RNA polymerase II (Pol II) binding regulated by Hairy in the blastoderm embryo using an inducible system as described previously to capture direct effects with high temporal resolution (*Li and Arnosti, 2011*) (*Figure 1A*). Hairy is first expressed in the *Drosophila* blastoderm embryo in a seven stripe pattern, which is important in controlling downstream pair rule genes that direct segmentation (*Ish-Horowicz and Pinchin, 1987*). Here, we express Hairy with a brief heatshock, throughout the embryo, which is sufficient to completely repress target genes such as *ftz* (*Figure 1A,B*). We treated the control embryos identically to embryos carrying the inducible Hairy transgene to test for possible nonspecific effects of heat shock on gene expression and chromatin marks. In this system, heat shock alone has no effect on the expression patterns of the pair rule and other genes analyzed, and the chromatin marks in heat shocked control embryos were indistinguishable from chromatin patterns previously reported for untreated embryos (*Li and Arnosti, 2011* and K Kok, data not shown). In total, we identified 241 down-regulated and 146 up-regulated transcripts in response to induction of Hairy (*Figure 1C*). Our microarray analysis captured previously identified targets of Hairy, showing downregulation of *en, edl, Impl2,* and *prd,* as well as *ftz,* all of which were previously found to be derepressed in *h* embryos (*Ish-Horowicz and Pinchin, 1987*; *Bianchi-Frias et al., 2004*).

Differentially regulated genes were compared to those physically bound by Hairy (*MacArthur et al., 2009*); 70% of down-regulated genes are bound by Hairy, suggesting that most of these are likely to be direct targets (*Figure 1C*). In contrast, only 30% of up-regulated genes are bound by Hairy, indicating that majority of these genes may be indirect targets. In situ hybridization and RT-qPCR confirmed the repression of a number of target genes we identified (*Figure 1B,D*). Many of these genes, including *odd, comm, comm2, edl, en, Impl2, prd,* and *18w,* have striped

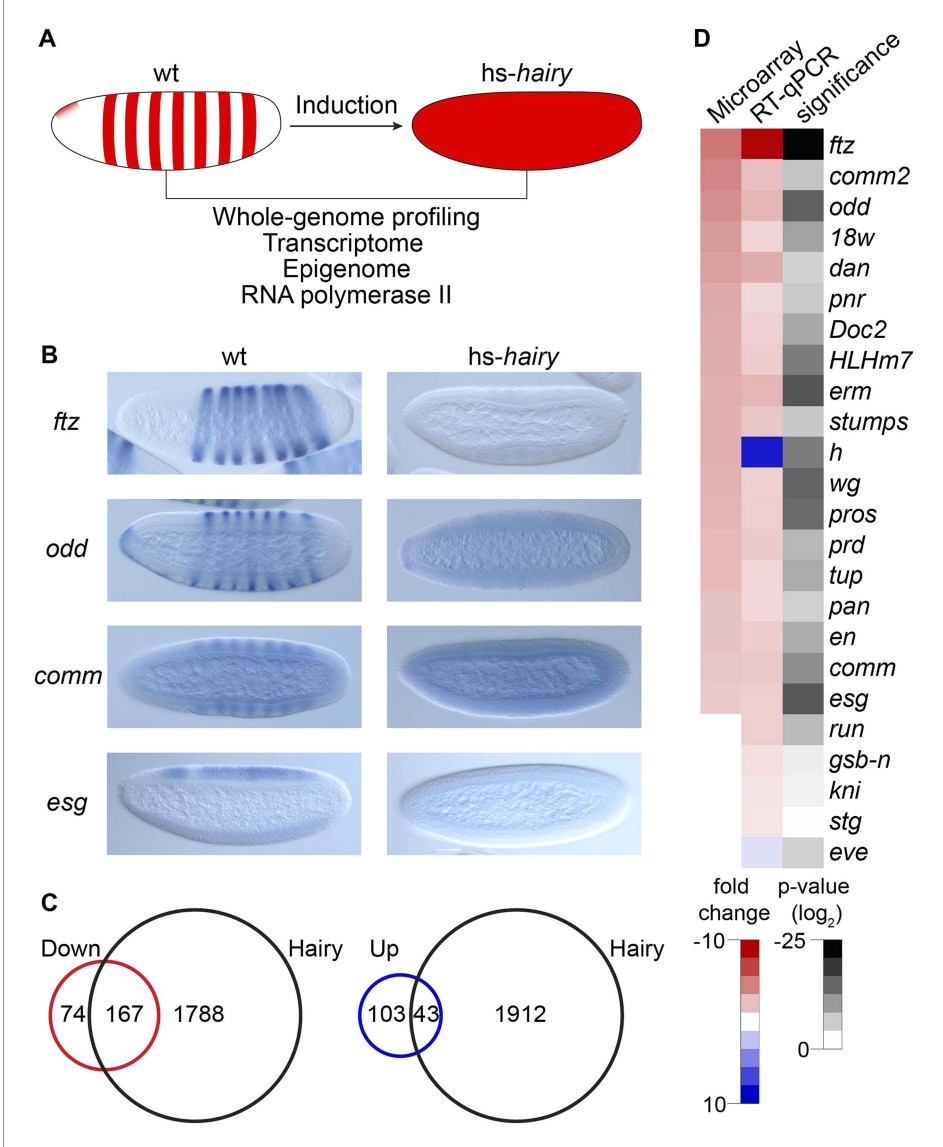

Figure 1. Global analysis of Hairy regulation. (A) Schematic expression of Drosophila embryo system used for Hairy repression, with outline of the genome-wide analysis of transcription, chromatin, and RNA polymerase II (Pol II). (B) Repression of *ftz*, *odd*, *comm* and *esg* revealed by in situ hybridization in wild-type (wt) and Hairy transgenic embryos (hs-*hairy*) after 20 min induction. Similar repression of *18w*, *HLHm7* and *erm* was also observed (not shown). (C) Transcriptionally regulated (red, down; blue, up) and Hairy bound genes identified by microarray and ChIP–chip (*MacArthur et al., 2009*). A larger fraction of down-regulated genes were physical targets of Hairy than for up-regulated genes (significance: p = 3.8e-95 and p = 3.5e-08 respectively, hyper-geometric test). Differentially expressed genes are selected based on p < 0.05 and fold change >2. (D) Validation of microarray data by RT-qPCR, showing concordance between these methods. Genes are ranked by the fold change from the microarray measurements. Significance was tested by Student's *t*-test. y-axis values were normalized as described in 'Materials and methods'.

The following figure supplement is available for figure 1:

**Figure supplement 1**. Similarity between binding of endogenous Hairy and overexpressed Hairy protein.

expression patterns complementary to that of Hairy, supporting direct regulation by the repressor. Furthermore, consistent with known biological functions of Hairy, gene ontology analysis showed that categories for down-regulated genes are significantly enriched in transcriptional regulation, cell fate commitment and neurogenesis (p < 3.7e-18). GO categories

for the set of upregulated genes were of lower statistical significance, and included reproductive processes (p < 0.03) (*Supplementary files 1, 2*).

Expression of the majority of genes bound by Hairy did not change (*Figure 1C*), consistent with previous observations that metazoan transcription factors have apparently many 'nonfunctional' interaction sites in the genome (*Cusanovich et al., 2014*).

## Coordinate chromatin transitions mediated by Hairy on diverse genes

Identification of functional and physical Hairy targets allowed us to study gene-specific chromatin changes associated with repression. We performed epigenomic profiling via chromatin immunoprecipitation-high throughput sequencing (ChIP-seq) of chromatin marks that are often correlated with specific features of *cis*-regulation; H4Ac, H3K27Ac, and H3K4me1 at promoters and enhancers; H3K4me3 at transcription start sites (TSS); H3K36me3 at gene body regions; and H3K9me3 at repressed regions of chromatin (*Zhou et al., 2011*). The measured signals for specific marks were highly reproducible in separate biological replicates, and Hairy-induced changes in histone marks were consistently observed at specific loci, such as the widespread loss of the H4Ac signal on the *ftz* locus, with little change to the overall global chromatin landscape (*Figure 2—figure supplement 1A*). As was apparent from comparison of control chromatin profiles, the induction of Hairy did not cause a global impact on histone marks. In the presence or absence of induced Hairy, the genome features for multiple chromatin marks are virtually identical, except in very discrete regions where there are significant changes (*Figure 2—figure supplement 1A–C*).

Using single gene techniques, we previously found that Hairy induces a widespread histone H4 deacetylation throughout the entire *ftz* locus (*Li and Arnosti, 2011*). To determine if these are general properties of Hairy, we compared all affected loci genome-wide. We observed that on a number of transcriptionally repressed target genes, H4 deacetylation is coupled with loss of the active marks H3K27Ac and H3K4me1. Widespread reduction of these active marks affecting >1 kb blocks was observed on many genes repressed by Hairy, including *ftz* and other segmentally expressed genes such as *h* and *18w* (*Figure 2*). Notably, Hairy regulates its own transcription by chromatin alteration, consistent with autoregulatory mechanism of related mammalian HES proteins (*Kageyama et al., 2007*). In addition to removal of enhancer marks, repression on *h* and *18w* resulted in demethylation of the promoter mark H3K4me3. Furthermore, action of Hairy on another pair rule gene, *odd,* was limited to removal of acetyl marks on H4 and H3K27; methylation marks on H3K4 are untouched (*Figure 2D*). These results suggest that Hairy mediates coordinated sets of chromatin transitions. The chromatin changes did however exhibit heterogeneous characteristics; the sizes of altered chromatin domains varied on different repressed genes. For example, changes in levels of H4Ac involved blocks with a range of sizes; generally larger than 1 kb, with the average ~2.5 kb. Somewhat smaller chromatin blocks were associated with repression of the *HLHm7, gogo, pros* and *tup* genes, which showed just as robust regulation of transcription as those genes with large tracts of chromatin modification (*Figure 2E–H*).

The largest ranges of size in chromatin domains were observed for H4Ac, but similar, although smaller ranges were also seen for H3K27Ac and H3K4me1 marks (*Figure 3*, *Figure 3—figure supplement 1A,B* and *Supplementary file 3*). We found strong correlations between the sizes of the domains of chromatin modification and the direct action of Hairy. Hairy-bound blocks of deacetylation were significantly larger than those not bound by Hairy, and smaller correlations were noted for other modifications, indicating that deacetylation is especially likely to show 'spreading' characteristics (*Figure 3*, *Figure 3—figure supplement 1A,B* and *Supplementary file 3*).

These results suggest that widespread effects found at H4Ac, H3K27Ac and H3K4me1 marks are dependent on presence of Hairy and are consistent with a long-range 'spreading' repression mechanism. We saw no correlation between the height or extent of Hairy binding sites and the range of chromatin alteration, suggesting that the effectiveness of this protein is not merely a function of number of binding sites (*Figure 3—figure supplement 2A,B*). Other local factors may dictate how extensively modifications are propagated on individual genes. Therefore, Hairy induces diverse chromatin transitions associated with gene silencing, indicating that there are gene-specific features dictating how repression is mediated at individual genes.

## Global set of chromatin modifications mediated genome-wide

These observations suggest there are context-specific aspects to chromatin modifications directed by Hairy. To determine the nature of changing chromatin states at different genomic loci, we compared

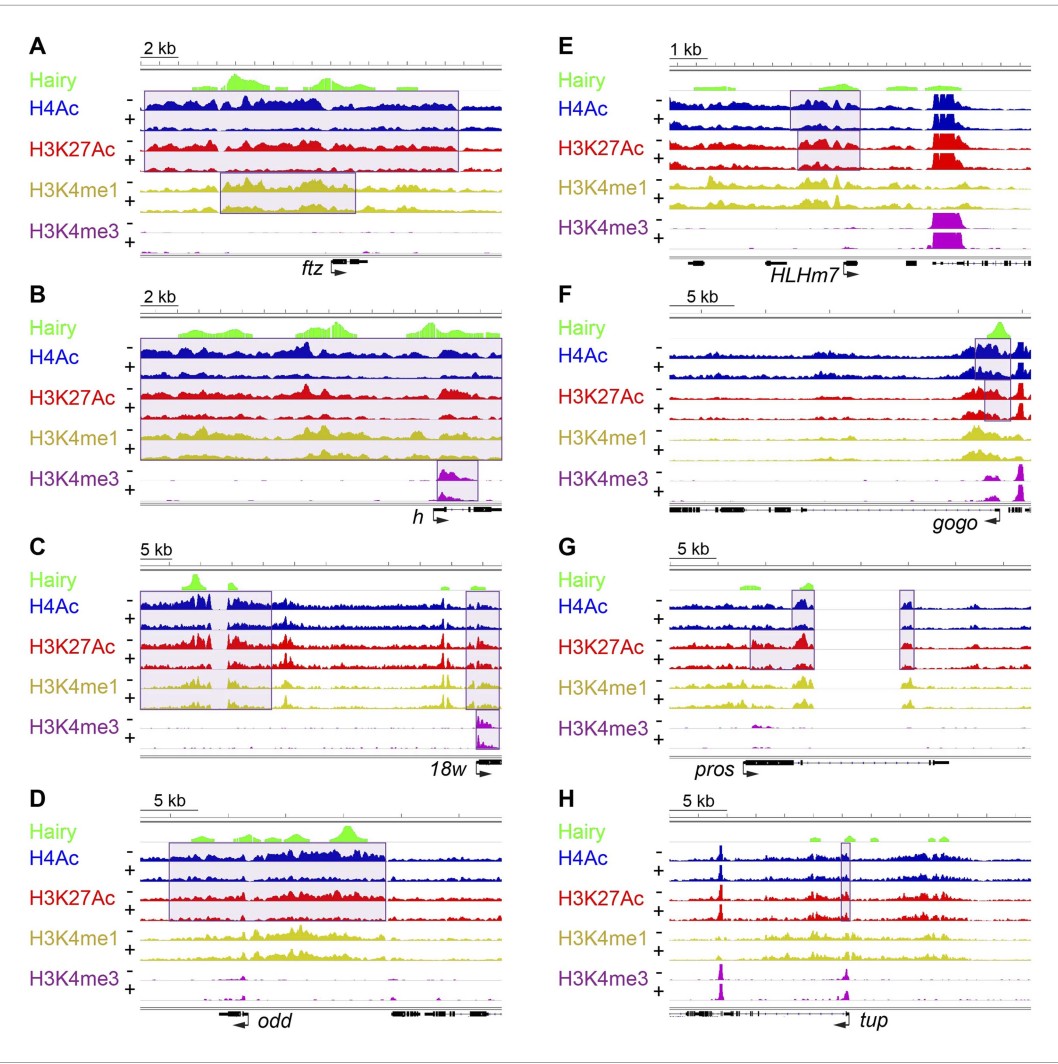

Figure 2. Examples of coupled, large-scale chromatin changes mediated by Hairy. Chromatin immunoprecipitation-high throughput sequencing (ChIP-seq) tracks for H4Ac, H3K27Ac, H3K4me1 and H3K4me3 are shown at repressed genes before (−) and after Hairy (+) induction, with gene models below. (A–D) Coupled reduction of active histone marks was observed in a wide-spread fashion on *ftz, h, 18w* and *odd* genes (scale at top left). (E–H) Relatively smaller blocks of chromatin changes were detected on *HLHm7, gogo, pros* and *tup* genes. Significantly changed regions (shaded boxes) were identified by the diffReps program. Hairy binding (top track) from *MacArthur et al. (2009)*.

The following figure supplement is available for figure 2:

**Figure supplement 1**. ChIP-seq reproducibility of biological replicates and variation between wild-type (wt) and transgenic embryos (H).

the complete set of significant alterations in all measured chromatin marks observed after Hairy induction, regardless of transcriptional effects on the neighboring genes. We observed both loss and gain of these marks on hundreds of regions. Most frequently observed were changes in H4Ac, H3K27Ac, H3K4me1 and H3K36me3; changes in some chromatin marks were much more frequent than in others, indicating that there is some heterogeneity in the impact of Hairy on different regions (*Figure 4A*). The changes in levels of these marks is not simply due to increased or decreased histone density, as histone H3 levels generally were unchanged (*Figure 4A*). The roughly equal abundance of regions showing loss or gain of acetylation and methylation would indicate that either secondary effects are common, or that Hairy may exert distinct biochemical activities on different loci. The correlation of Hairy-bound regions with repressed transcripts, as well as the association of Hairy binding

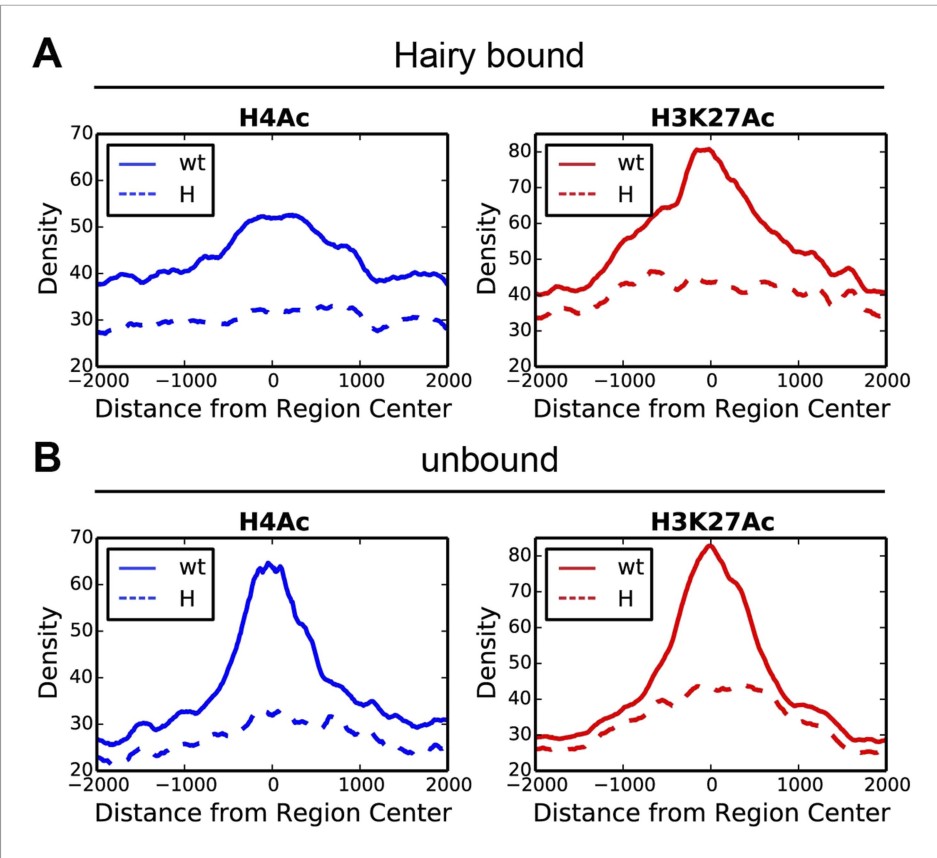

**Figure 3**. Direct Hairy target genes exhibit broad domains of chromatin effects. Distribution of genome-averaged ChIP-seq signals before (straight line) and after (dashed line) Hairy induction, showing 4 kb window around affected regions. (**A**) Distributions of histone H4Ac and H3K27Ac marks of direct Hairy targets were significantly broader than for regions (**B**) not bound by Hairy (p = 2.55e-92 and p = 5.63e-70 respectively; KM test).

The following figure supplements are available for figure 3:

**Figure supplement 1**. Distinct chromatin profiles associated with direct and indirect Hairy targeted loci.

**Figure supplement 2**. Little correlation between height of Hairy peaks or width of Hairy-bound region and extent of H4 deacetylation blocks and width (**A**) or height (**B**) of Hairy peaks.

with longer-range deacetylations, but not with increased acetylation, supports the idea that indirect effects are common. Indeed, focusing specifically on genes targeted by Hairy, we found that H4 histone deactylation was strongly enriched compared to acetylation gains, suggesting that deacetylations are direct effects (*Figure 4B* and *Figure 4—figure supplement 1A*). Further support comes from consideration of the actual Hairy occupancy of the chromatin blocks in question; there was significant correlation between Hairy binding and chromatin blocks exhibiting decreased, but not increased acetylation (*Figure 4—figure supplement 2*).

With respect to another chromatin mark, changes in histone methylation revealed an unexpected and interesting trend. Both decreases and increases in H3K4me1 signals were significantly associated with Hairy-bound genes; decreases were especially found in those regions directly bound by Hairy (*Figure 4—figure supplement 1A*, *Figure 4—figure supplement 2* and *Figure 4B*). At the same time, about one-quarter of the genes that were transcriptionally silenced by Hairy showed increases in H3K4me1, although these regions of increase did not overlap with Hairy binding. The increase in this mark may represent a reaction of proximal promoter chromatin to distal enhancer silenced by Hairy.

H3K36me3 modification is often associated with active transcription. We found a small fraction of transcriptionally regulated genes that exhibited changes in the mark upon transcriptional repression

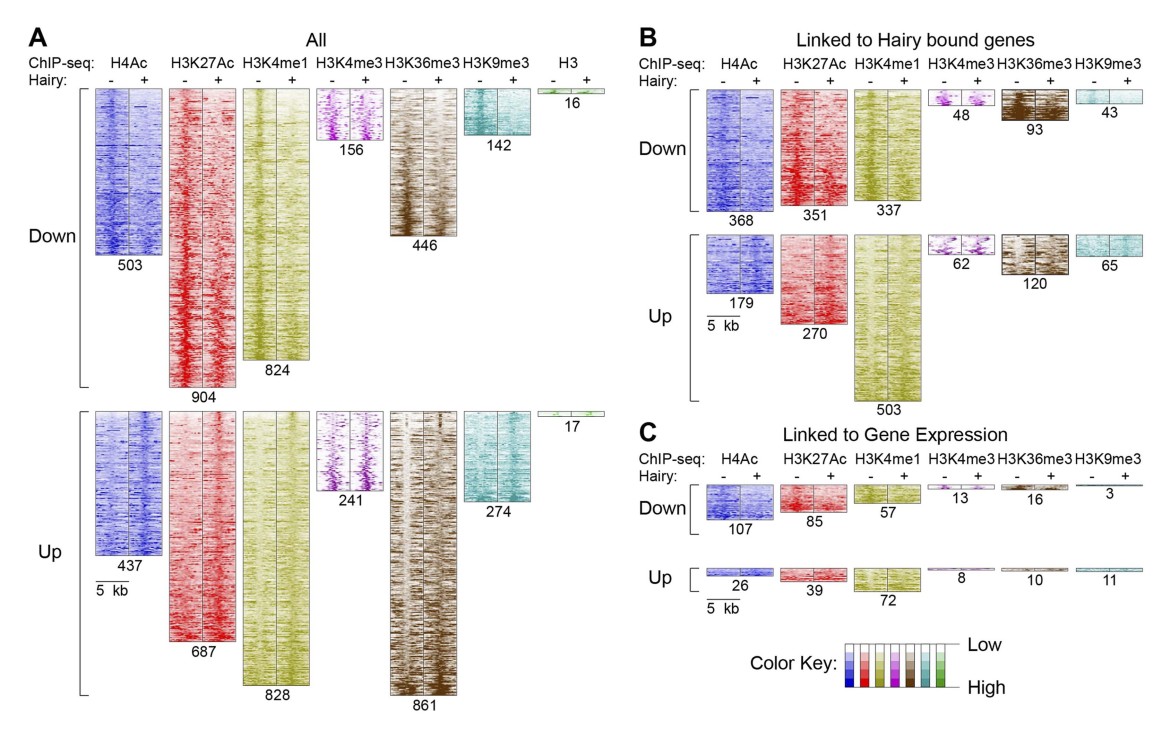

**Figure 4**. Pervasive genome-wide chromatin effects of Hairy. (**A**) All reduced (top) and increased (bottom) chromatin marks in the genome for H4Ac, H3K27Ac, H3K4me1, H3K4me3, H3K36me3, H3K9me3 and H3 shown as heatmaps for 5 kb windows from the center of significantly affected regions before (−) and after (+) Hairy induction. The number of affected regions indicated below each mark. (**B**) Affected chromatin regions associated with Hairy-bound genes show preferential enrichments for H4Ac, H3K27Ac, and H3K4me1. All affected regions were assigned to closest genes, and those in the vicinity of Hairy-bound genes are shown. (**C**) Subset of modified regions from (**B**) that were linked to genes transcriptionally regulated by Hairy. Significance of enrichment for chromatin modifications shown in *Figure 4—figure supplement 1A,B*.

The following figure supplements are available for figure 4:

**Figure supplement 1**. Significance of individual histone modifications associated with Hairy bound genes and transcriptionally regulated genes.

**Figure supplement 2**. Strong correlation between the presence of Hairy binding and chromatin alterations on specific chromatin blocks.

(*Figure 4C* and *Figure 4—figure supplement 1B*). These findings indicate that Hairy repression does not require H3K36me3 changes. Indeed, the direct effect of H3K36me3 on transcription is complex, as has been found for many other histone marks. For example, upregulation of KDM4A histone demethylase target genes in *Drosophila* occurs without increases in H3K36me3 (*Crona et al., 2013*). Similar studies with elongation factor Spt6 in *Drosophila* further indicate that *Hsp70* gene expression is not correlated to H3K36me3 levels (*Ardehali et al., 2009*). In fact, H3K36me3 may in some contexts contribute to gene silencing, due to its presence in heterochromatic domains (*Chantalat et al., 2011*) and in other cases, removal of H3K36me3 is required to promote transcriptional elongation (*Kim and Buratowski, 2007*).

A smaller number of H3K9me3 regions were observed to change globally, or on genes that were associated with Hairy (*Figure 4A,B*). Very few repressed genes showed any alteration in this mark, thus it appears that repression mediated by Hairy does not require changes in such repressive histone modifications (*Figure 4C*), consistent with our previous report that repression on *ftz* did not change H3K27me3 levels (*Li and Arnosti, 2011*). Indeed, other studies have found that these marks are not always simply coupled to repression. For example, only a modest correlation between H3K9me3 and H3K27me3 levels and gene silencing was observed in human cells (*Barski et al., 2007*; *Zhang et al., 2012*). In the differentiation of T and B cells, only a small fraction of repressed genes ever acquire H3K27me3 (*McManus et al., 2011*; *Zhang et al., 2012*). Interestingly, H3K9me3 was found to be

enriched in many active promoters and associated with transcriptional elongation in vertebrates (*Vakoc et al., 2005*; *Squazzo et al., 2006*).

Consequently, of the assessed modifications, it appears that Hairy predominantly works to modify acetyl and methyl marks of H4, H3K27 and H3K4 and represses gene expression primarily by eliminating active marks.

Of all chromatin regions impacted by Hairy, only a small number are associated with genes demonstrating measurable transcriptional changes (*Figure 4C*). Thus, it is striking that the majority of chromatin changes are decoupled from any detectable effect on gene expression (*Figure 4—figure supplement 1B*). For the many cases where chromatin effect was unlinked to changed gene expression, we observed extensive chromatin alterations associated with both silent and active genes. For example, chromatin transitions occur on transcribed genes not functionally repressed by Hairy, as seen on the *pyr* gene (*Figure 5A*). In this case, the gene may remain active because the necessary *cis*-regulatory elements are located distally and are still able to interact with the promoter and activate it. In other cases, chromatin changes flank silent loci; *nht* undergoes widespread deacetylation and demethylation even though it is silent during this developmental stage of embryos (*Figure 5B*). In some cases, binding and changing chromatin near inactive genes by Hairy in the blastoderm embryo may involve the interaction of Hairy with DNA elements that will become active at a later developmental stage, however, this seems unlikely in the case of *nht*, a testes-specific gene. Here, the physical binding by Hairy and subsequent impact on chromatin may represent 'errant targeting'. Overall, chromatin changes were observed to correlate with over half of the regions bound by Hairy, suggesting that in most cases, this protein is biochemically active on chromatin, whether or not the changes lead directly to gene repression (*Figure 5—figure supplement 1*).

## Hairy coordinates sets of modifications on preferred gene regions

The individual cases described in *Figure 2* suggest that Hairy organizes a coordinated set of chromatin changes involving both deacetylation and demethylation of multiple histone residues. To determine if such alterations are a general property of the repressor, we assessed the extent of coordination of modifications on all individual blocks of affected chromatin. Changes in H4Ac, H3K27Ac and H3K4me1 marks were significantly correlated at many loci (*Figure 6A*). Deacetylation events were also strongly correlated with loss of both H3K4me1 and H3K4me3, indicating that Hairy may form complexes containing both deacetylase and demethylase activities. Indeed, the CtBP cofactor is known to bind both of these classes of enzymes. However, Hairy is not mediating only one average type of transformation; removal of methyl groups from H3K4me1 and H3K4me3 is catalyzed by distinct classes of enzymes; Hairy is likely to interact with both, allowing for removal of H3K4me1 marks on distal sites and H3K4me3 at TSS (*Figure 6B*). A very similar pattern of correlations between acetylation marks, and between acetylation and methylation marks was observed for regions with increased acetylation and methylation.

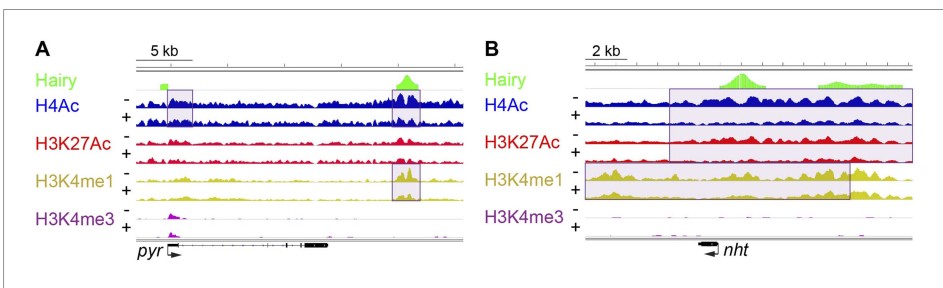

**Figure 5**. Examples of chromatin-modified loci unlinked to changes in gene expression. (**A**) *pyr* is actively transcribed, and not significantly repressed by Hairy, (**B**) while *nht* is not expressed at this stage. ChIP-seq tracks for H4Ac, H3K27Ac, H3K4me1 and H3K4m3 are shown before (–) and after Hairy (+) induction.

The following figure supplement is available for figure 5:

**Figure supplement 1**. Global association of Hairy binding with histone mark alterations.

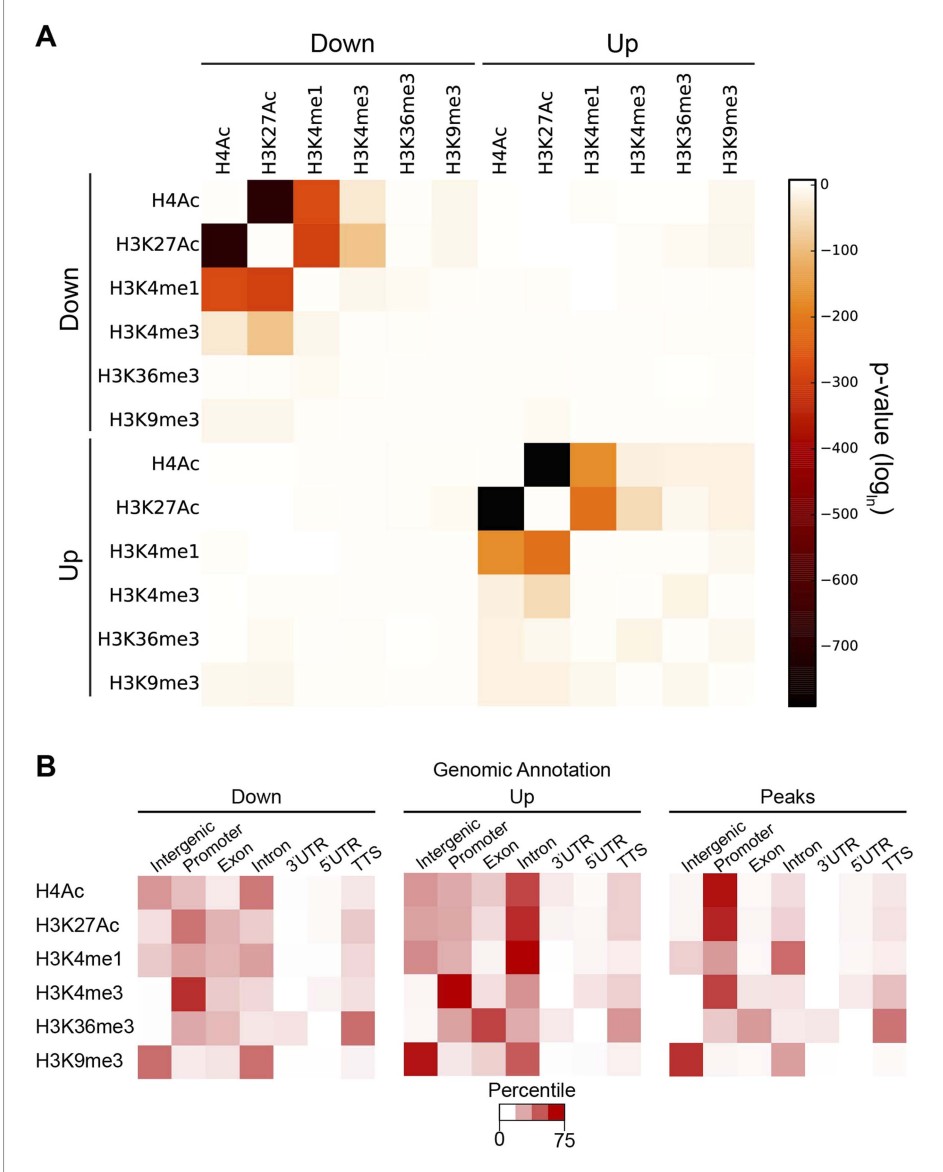

**Figure 6**. Coordination in changes of specific chromatin modifications by Hairy. (**A**) Very strong overlap between decreases in regions of H4Ac, H3K27Ac, H3K4me1 (heat map, upper left quadrant). Similar coordination between increases of H4Ac, H3K27Ac, H3K4me1 was noted (lower right quadrant). Combined increases and decreases of different marks were rarely observed. (**B**) Distribution of modified blocks by genomic regions show preferential action of Hairy at a distance from transcription start site (TSS). Affected regions were mapped to intergenic regions, promoter, exon etc. Overall distribution of genomic peaks for measured marks shown at right; the distributions for affected H4Ac and H3K27Ac regions deviated from the genomic averages (left, decreased, and center, increased levels).

These elements may represent to a large extent indirect targets of Hairy, as no significant overlap between Hairy binding and these modified regions was found (*Figure 4—figure supplement 2*).

Where does Hairy most commonly mediate significant chromatin modifications? We compared the location of individual histone marks genome-wide to those altered by Hairy expression. Although a third of Hairy binding sites are promoter-proximal, where the majority of H4 and H3K27 acetylation occurs, the large majority of affected chromatin sites were found on intergenic and intronic regions, suggesting that successful alterations are targeted to distal sites that may represent transcriptional enhancers (*Figure 6B*). By contrast, changes in the methylation marks H3K4me1,

H3K4me3, H3K36me3, and H3K9me3 are found in the genomic regions where they are naturally enriched (*Figure 6B*). For instance, H3K4me3 marks are enriched at TSS, as are the bulk of the altered chromatin sites. Hairy may thus have privileged sites on which it is more likely to induce chromatin changes; promoter regions may be in general more resistant to acetylation changes if strong activators are replenishing acetylation marks at these loci. In addition, transcriptional targets of Hairy are enriched in developmentally regulated genes, which typically possess larger *cis*-regulatory regions with multiple distal enhancers (*Supplementary file 1*) (*Nelson et al., 2004*).

## RNA Pol II and silencing by Hairy

To directly assess the influence of Hairy on transcriptional machinery, we compared the genome-wide occupancy of RNA Pol II before and after Hairy induction. 75 of 241 repressed genes exhibited changes in Pol II occupancy (*Figure 7A*). Only three of those are not directly bound by Hairy, indicating a direct regulation by Hairy in loss of Pol II signal. A marked decrease of Pol II occupancy was observed

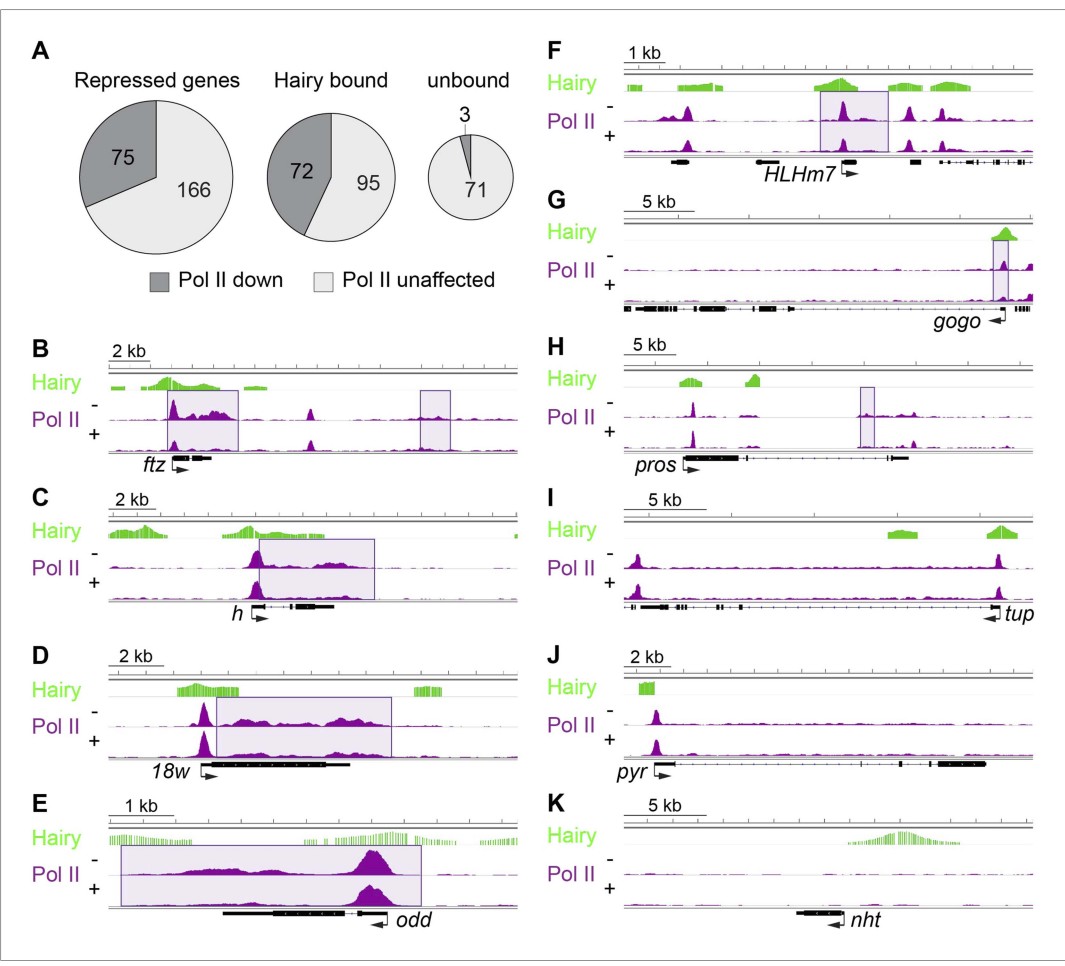

**Figure 7**. Diverse impact on RNA Pol II occupancy by Hairy. (**A**) A minority of genes show significant changes in Pol II occupancy after Hairy repression, although a larger proportion of the directly targeted genes have measureable decreases in Pol II. 'Repressed genes' shows entire set of transcriptionally downregulated genes, with reduced Pol II occupancy shown in dark gray. Subsets of genes directly bound or not bound by Hairy shown in center and at right. (**B–I**) Pol II occupancy on transcriptionally regulated genes before (−) and after (+) Hairy induction. Pol II occupancy decreases in the promoter and gene body of *ftz* and *odd*, only on the gene body of *h, 18w* and *pros*, and only at the promoter of *HLHm7* and *gogo*. Pol II signal was not changed significantly on *tup*. (**J, K**) Consistent with lack of transcriptional effects on other genes with associated chromatin modifications, Pol II occupancy on *pyr* is not changed, and absent on *nht*.

at the *ftz* promoter, gene body and distal downstream region (*Figure 7B*). Loss of binding at the promoter, or the body of the gene, or both was detected on other loci (*Figure 7C–I*). Thus, the loss of Pol II on the promoter and gene body of *ftz* is not universally associated with transcriptional repression; on other genes, silencing of a distal enhancer may interfere with promoter release without blocking polymerase recruitment to the promoter, consistent with recent studies implicating transcriptional signaling in promoter escape, rather than promoter recruiting (*Lagha et al., 2013*). As expected, genes with associated chromatin changes without any impact on transcription did not show any change on Pol II occupancy (*Figure 7J,K*).

95 repressed genes bound by Hairy did not show any change in Pol II occupancy (*Figure 7A*). It is possible that Hairy induces a slower transit rate of Pol II without any detectable change in Pol II binding. It has been suggested that repression through elongation control may cause no change in Pol II binding on *slp1* and *Hsp70* (*Adelman et al., 2006*; *Wang et al., 2007*; *Ardehali et al., 2009*). Our previous analysis of *eve* repression by short-range repressor Knirps showed similar effects (*Li and Arnosti, 2011*). Therefore, Hairy may interfere with gene expression at different steps of the transcription cycle, as also suggested for repression by the glucocorticoid receptor, indicating gene specific repression mechanisms (*Gupte et al., 2013*). An additional consideration is that genes featuring poised polymerase at the promoter in many or most nuclei, but are only expressed in a few nuclei, will have weak signals at the body of the gene. Therefore, the lack of change in Pol II levels on the gene body would reflect the inherently low signal, rather than a distinct biochemical mechanism. This explanation may account for a considerable number of affected genes where no changes in Pol II levels are observed after repression.

## Predicting a 'successful' repression context

The complexity of chromatin transitions observed genome-wide in the wake of Hairy expression prompted us to ask which features best predict successful repression of a target gene, vs those genes with no chromatin responses or exhibiting errant targeting by Hairy. Here, we alter the expression of only one regulatory factor, rather than the many changes in regulatory factors observed over a developmental time course, therefore our data sets are enriched for direct action of Hairy, potentially simplifying the search space. We sought out correlations between dynamic histone marks, Pol II, Hairy, CtBP and Gro and the repression of targeted genes. Direct inspection reveals that occupancy by Hairy, Gro, and decreases in Pol II are strong predictors of repression, as are several histone marks, compared to genes unaffected or those activated (*Figure 8A*). However, there are numerous loci that do not fit these simple generalizations. To more systematically assess the connections between these different observed states and transcriptional repression, we applied machine learning to analyze features that may be implicated in the activity of Hairy. We tested 41 features, including the number of observed peaks for Hairy, CtBP, and Gro; the number, width, and magnitude of altered chromatin blocks, and distance to TSS for 583 genes (241 repressed, 146 activated and 196 unaffected genes; activated and unaffected genes were grouped as nonrepressed genes). To identify the most informative features, four different feature selection algorithms were used to rank the information content of the 41 measured properties associated with the genes; the top twenty of these features were then used for predictions (*Supplementary file 4*). We then tested four classifiers, using 90% of the data for training and 10% for predictions, with 10-fold cross-validation. Overall, each of the classifiers performed better than background, with Random Forests showing superior performance of ~75% accuracy for repressed and nonrepressed genes (*Figure 8B*). Three of the feature selection algorithms used with this classifier employed very similar features to achieve this high level of accuracy (*Supplementary file 4*), indicating that certain features are most informative. The presence and properties of Hairy and Gro peaks are good indicators, although not sufficient information by themselves. RNA Pol II properties, transcript levels, and chromatin modifications, especially H3K4me1 and H4Ac, whether causal or not, are also a close reporter of gene activity. The overall performance differences in these methods are frequently observed in machine learning studies, and likely reflect the underlying data structure and types of features available for analysis. Genes that were correctly predicted as repression targets generally had the most differential features, including binding by Hairy and Gro, and changes in histone modifications. The genes that were least successfully called had one or no differential features, and may represent genes that are expressed in fewer cells and at lower levels where measurement of chromatin changes in a global population is difficult (*Figure 8C*). The nonrepressed gene *pyr* was consistently called as 'repressed' by the machine learning algorithms, as it exhibited chromatin signatures similar to those

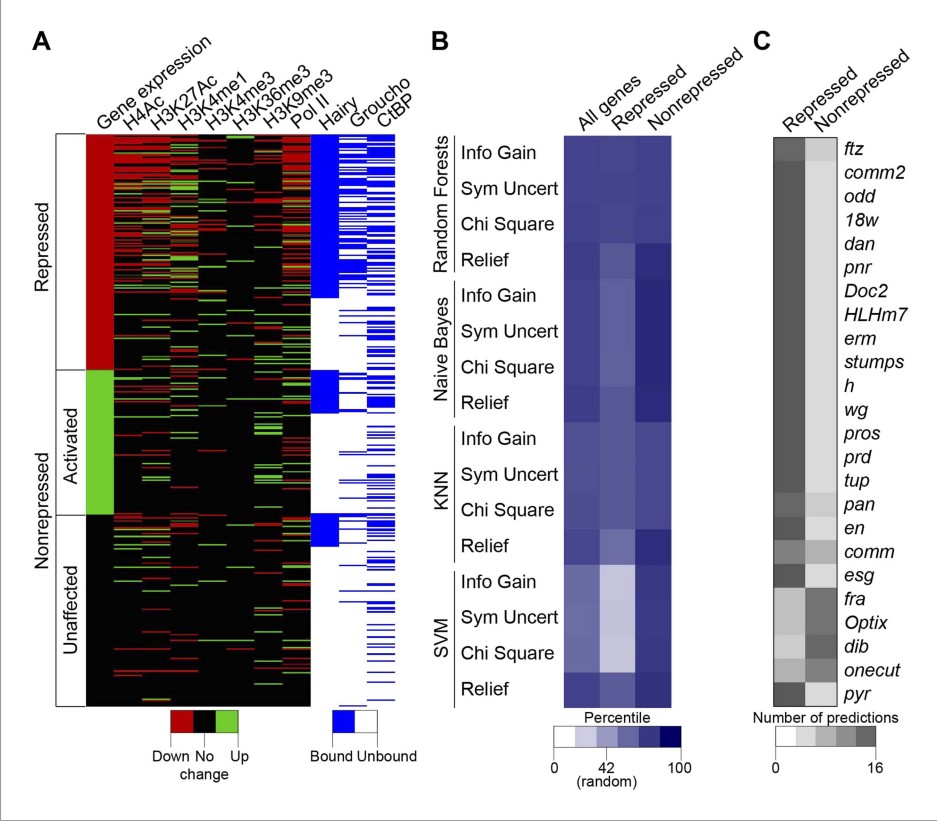

**Figure 8**. Machine learning reveals complex chromatin code for repression of Hairy target genes. (**A**) Changes in histone marks, Pol II occupancy and Hairy, Gro and CtBP binding on repressed (red), activated (green) and unaffected (black) genes upon Hairy induction. Genes were grouped by change in expression, then subgrouped into Hairy bound or unbound, and finally ranked by fold change in gene expression. Activated and unaffected genes were grouped as nonrepressed genes. (**B**) Relative success rate at calling repressed and nonrepressed genes for four different machine learning models. Background prediction for this entire set is expected to be 58%; Random Forests, Naive Bayes, KNN classifiers had an average success of 75% overall, while the SVM classifier was not better than background. Classifiers were used in conjunction with Information Gain, Symmetrical Uncertainty, Chi Square and Relief feature selection algorithms. The average prediction accuracies of each method are shown in the first column. Expected random success (42%) for repressed genes (middle column) shown on heat map scale bar. (**C**) Model predictions for subset of repressed genes including those identified in *Figure 1*; top 19 were successfully predicted by almost all methods. *fra*, *Optix*, *dib*, and *onecut* were genes with disparate predictions that had few measureable chromatin features. At bottom, uniform false 'repressed' calls for *pyr*, which was not transcriptionally repressed.

found on genes that were actually repressed (*Figure 8C*). In this case, we propose that the relevant enhancers lie outside of the chromatin regions affected by Hairy. Such genes may represent loci that are poised for capture in the Hairy regulatory network through stepwise acquisition of activator binding sites. Overall, this analysis indicates that from the perspective of Hairy biochemistry, there are intuitive and some non-intuitive combinations of chromatin dynamics that typify this protein's action in the context of transcriptional repression, rather than a 'practice' site, but other factors predominate in many instances. The missing information likely relates to the activity of bona fide *cis-* regulatory elements that are acting on genes in the vicinity of Hairy, which is partially but incompletely known from genome-wide studies (*Kvon et al., 2014*).

## Discussion

By testing direct effects of the Hairy repressor in the embryo, we conclude that this protein coordinates a stereotypical set of chromatin modifications, modulated by local context, that underlie

its function as a long-range repressor. Most remarkably, these changes on chromatin impact large segments of the genome that are not directly relevant to gene expression in this developmental context. We speculate that these off-target activities may provide an easy entry point for evolution of novel regulatory switches (*Figure 9*). Our mechanistic analysis of Hairy provides insights into likely mechanisms of related HES factors, as well as other transcriptional repressors that serve as scaffolds for chromatin modifying complexes. Hairy interacts with the widely utilized cofactors Gro, CtBP, and the Sir2 HDAC, and here we provide for the first time a genome-wide picture of the biochemical activities of this archetypal repressor.

How is transcription actually controlled by Hairy? The associated chromatin modifications may be effects, rather than direct causes of gene silencing. Our previous studies indicated that Hairy modulated transcription independent of activator occupancy or SAGA co-activator occupancy (*Martinez and Arnosti, 2008*). These previous observations raised the possibility that Hairy acts through entirely independent pathways from that employed by activators to block transcription. Our work here indicates that Hairy does indeed directly reverse chromatin marks associated with activators, and may therefore work through a dynamic competition with these activators, undoing their positive influence on the chromatin environment that would be necessary for RNA polymerase initiation and/or elongation (*Figure 9A*). Indeed, Hairy repression is readily reversible, with genes showing reversion to an active state minutes after depletion of the overexpressed repressor (K Kok, unpublished results).

The genome-wide analysis of repression by Hairy revealed an unexpected facet of chromatin activity and highlights the need to consider the activity of 'off target' sites in generating novel elements, particularly because for Hairy at least (and likely other factors that employ the same cellular machinery) they are 'shovel ready' and not constrained by complex *cis*-regulatory grammar. Metazoan transcription factors typically interact with thousands of discrete sites in the genome, but only a small subset of these interactions correlate with observable effects on gene expression. In this study, we combined analysis of gene expression and chromatin dynamics in a way that allowed us to attribute effects directly to the induction of Hairy, inferences that would be difficult with a loss-of-function assay due to kinetics of depletion and secondary effects. In contrast, many other genome-wide data sets provide a static snapshot of the extant chromatin landscape or track complex changes through development, which represents the combined contributions of many activators and repressors. Previous studies have noted the presence of detectable but lowly-occupied sites, which have been suggested to reflect non-specific, non-functional interactions that are unavoidable by-products of proteins binding to large genomes (*Fisher et al., 2012*). Other studies have emphasized that

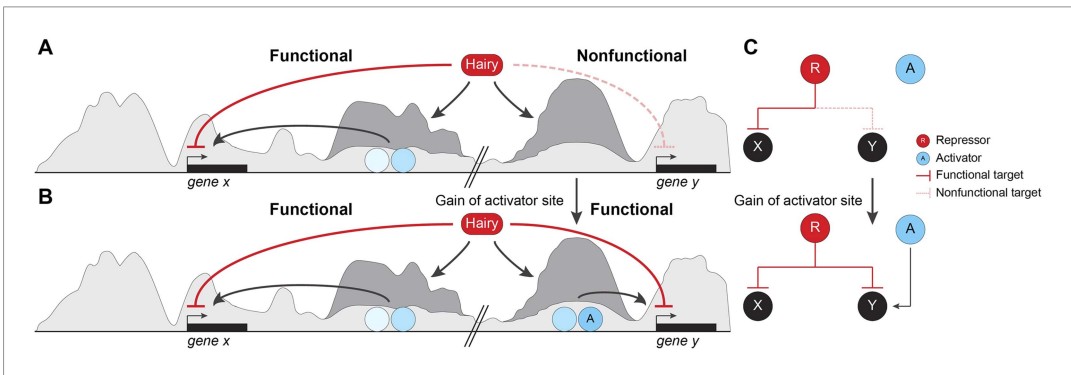

**Figure 9**. Pervasive biochemical activities on 'off-target' loci may represent molecular exaptations that generate novel edges between nodes of a standing gene regulatory networks (GRN). Functional and nonfunctional interactions of Hairy with chromatin are depicted. (**A**) Hairy repression of target genes results in loss of active histone marks such as H4Ac, H3K27Ac, and H3K4me1 (dark gray peaks; *gene x*). Hairy interacts with many other nonfunctional targets where it carries out biochemical activities similar to those seen on transcriptionally controlled loci (*gene y*). The latter chromatin changes are inconsequential and unlikely to be evolutionarily selected. (**B**) Gain of activator sites in a region of Hairy-modified chromatin may generate an on/off switch and result in functional targeting. (**C**) Schematic representation of cooption of Hairy physical interaction into modified GRN.

transcription factors may have general nonspecific interaction with HOT sites that represent preferences for open chromatin (*Gerstein et al., 2010*; *modENCODE Consortium et al., 2010*). In general, the overall view is that whether or not these interactions are conserved, they may be of little functional consequence, and are not important for activity of GRNs (*Cusanovich et al., 2014*). Importantly, considering our finding that 'off-target' Hairy sites still appear to regulate chromatin structure, we should fundamentally reconsider how we interpret genome-wide data sets. Frequently, an increase in H3K27 acetylation is taken as an indication that the element is an active enhancer, without further functional tests (e.g., *Villar et al., 2015*). Of course, correlated gene expression measurements indicate that such elements are likely to be enhancers in many cases, but genomic consideration of chromatin marking must not automatically equate changes in certain active marks with enhancers.

Our study provides a new perspective on these previous observations, in that essentially trivial biological interactions may have consequences in evolutionary time. We show that Hairy is engaged apparently in errant targeting of chromatin on many loci during the period when it is expressed, and demonstrate that in many cases, little distinguishes the types of chromatin effects observed on functionally repressed targets compared to 'non-functional' interactions on other loci (*Figure 9A,B*). Thus, unlike an earlier model for Hairy action, in which the protein is active only when embedded in a previously active enhancer (*Nibu et al., 2001*), our work demonstrates that Hairy is able to mediate biochemical activities in most bound regions, indicating that there is little context necessary for the protein to function. Therefore, Hairy may be relatively nonselective about where it can attract chromatin-modifying agents across the genome. Much molecular biology research has emphasized the high degree of cooperativity necessary for metazoan transcription factors to work well. Enhanceosomes, patterning elements and other enhancers give aberrant readouts if correct stoichiometries and spacings are not respected. These findings suggest that random individual sites are less likely to generate a suitable transcriptional readout. At least for repressors such as Hairy, the demands for generating biochemical activity are lower than anticipated, indicating that enhancers may have a lower threshold for formation that we might have expected. Although some of the targeted genes that are not transcriptionally affected may represent ectopic binding events of the induced Hairy protein, most sites are found in ChIP analysis of endogenous Hairy. The unresponsive genes may in some cases represent later targets of Hairy, may be already repressed by endogenous Hairy, or may have responses too small to measure in this system, however it is likely that there are hundreds of changed chromatin regions that not formally part of the functional Hairy GRN. Thus, a large fraction of the genomic interactions are likely to be with regions that are not strongly selected on an evolutionary timescale. As long as the induced chromatin changes are inconsequential, these effects will not be selected against during genomic evolution. This biochemical activity, however, may provide a unique molecular exaptation to generate novel edges between nodes of a standing GRN (*Figure 9B*). Most enhancers involve the combined action of transcriptional activators and repressors, thus errant targeting may facilitate formation of new modules with gain of a few activator binding sites (*Gould and Vrba, 1982*) (*Figure 9C*).

## Materials and methods

### Plasmid construction

The heat-inducible *hairy* gene was created by introducing a multiple cloning site containing Kozak sequence, initiator ATG and HindIII/BglII sites into the 5′ portion of the *hairy* ORF in the pCaSpeR-hsh using EcoRI/BstEII sites as described previously (*Li and Arnosti, 2011*). 400 bp of upstream promoter, 5′ UTR, Kozak sequence, initiator ATG, HindIII/BglII sites, coding sequence and entire *hsp70* 3′ UTR from the modified pCaSpeR-hsh were amplified using 5′ and 3′ primers with AgeI/KpnI sites and subcloned to the modified pattB vector (*Sayal et al., 2011*). Oligonucleotides with sequence encoding the double Flag epitope, as described in *Zhang and Arnosti (2011)*, was inserted 5′ of the coding sequence after the ATG using HindII/BglII sites, so that Hairy protein was expressed with the double Flag tag at the N terminus.

### Embryo collection, in situ hybridization and antibody staining of *Drosophila* embryos

For chromatin analysis 2–3.5 hr embryos were collected and 20 min heat-shock treated for induction of transgenes as described previously (*Li and Arnosti, 2011*). We treated the wild-type embryos similar to embryos carrying inducible transgene to control for possible nonspecific effects of heat

shock. Heat shock alone has no effect on the expression or chromatin patterns (data not shown). For analysis of gene expression by in situ hybridization, embryos were fixed and stained using anti-digoxigenin-UTP-labeled RNA probe for *ftz* as described previously (*Struffi, 2004*).

## Quantitative reverse transcriptase PCR analysis

Total RNA from embryos was purified using RNeasy columns (Qiagen), and reversed transcribed using a High Capacity cDNA Reverse Transcription Kit from Invitrogen/Applied Biosystems. The cDNA was then analyzed by real-time PCR using the primer pairs located at transcription units. Data was normalized to *act5c*. Values for wild-type embryos were set to 1; results represent the average of 2–8 biological replicates. Statistical significance was tested using Student's *t*-test and p < 0.05. Amplicons were designed using Primer Express and Primer-BLAST.

## Expression profiling analysis

Total RNA from 2–3 hr embryos was purified using RNeasy columns (Qiagen, Valencia, CA). Samples were amplified and labeled using the Quick AMP Labeling kit (Agilent, Santa Clara, CA) and hybridized to 8 × 15K Customized *Drosophila* Genome Oligo Microarrays (Agilent) according to the manufacturer's instructions. Slide image data was quantified using Agilent's Feature Extraction software. Four biological replicates were performed for each sample. Differential gene expression analysis was performed with the GeneSpring program (Agilent). Functional annotation of down- and up-regulated genes was done using the Database for Annotation, Visualization and Integrated Discovery (*Dennis et al., 2003*). Differentially regulated gene symbols and their fold changes are listed in *Supplementary file 5*.

## Chromatin immunoprecipitation

Heat shocks and ChIPs were performed as described previously (*Li and Arnosti, 2011*), with the exceptions that embryos were sonicated for a total of 20 times using a Branson sonicator in 1 ml of sonication buffer. After precipitation of chromatin-antibody complexes, protein A beads were washed twice with low-salt buffer, once with high-salt buffer, once with LiCl buffer and twice with Tris-EDTA. We used the following antibodies: rabbit IgG (5 µl, Santa Cruz Biotechnology), rabbit anti-H3 (1 µl, Abcam, Cambridge, MA), rabbit anti-acetyl H4 (1 µl, Upstate, EMD Millipore, Billarica, MA), rabbit anti-acetyl H3K27 (1 µl, Abcam), rabbit anti-monomethyl H3K4 (1 µl, Abcam), rabbit anti-trimethyl H3K4 (1 µl, Abcam), rabbit anti-trimethyl H3K36 (2 µl, Abcam), rabbit anti-trimethyl H3K9 (3 µl, Abcam), rabbit anti-Flag (5 µl, Sigma-Aldrich, St. Louis, MO), rabbit anti-Rpb3 (5 µl, gift from Carla Margulies, LMU University of Munich).

## ChIP-seq

### Libraries

DNA from chromatin immunoprecipitation (10 ng) was adapter-ligated and PCR amplified (18 cycles) as described in *Ford et al. (2014)*. DNA ligated to the adapter was size selected for 300–500 bp. Illumina HiSeq single-end reads were checked using FastQC and HOMER for sequence quality, base sequence and GC content, sequence duplication, sequence bias, overrepresented sequences and Kmer content. Reads were aligned to genome (BDGP 5.70) with Bowtie version 1.0.0 using–m 1–best parameters. Tags that only mapped uniquely to the genome were considered for further analysis. Summary of tags generated is shown on *Supplementary file 6*. ChIP-Seq experiments were visualized as custom tracks using Integrative Genomics Viewer (Broad Institute). Total uniquely mapped tags were normalized to 10 million reads to generate tracks. y-axis values shown in all figures use the same scale for an individual measurement of each histone modification in the individual panels. For reasons of clarity, scales can vary between different panels.

### Mapping differential regions

We detected the regions where chromatin states are changed upon induction of Hairy by comparing the level of histone marks at particular genomic locations. Differentially changed genomic regions were identified using the diffReps program (*Shen et al., 2013*), which uses a sliding window approach to scan the genome and find regions showing read count differences. Default window size with–nsd broad–meth nb parameters was used for the analysis. For downstream analysis, we used regions with

p < 0.05 and fold change (log2) > 0.4 or fold change (log2) < −0.4. Input was sequenced from nontransgenic (wt) and Hairy overexpressing embryos and used as background.

Hypergeometric Optimization of Motif EnRichment (HOMER) was used for peak finding and downstream data analysis (*Heinz et al., 2010*).

## Identification of ChIP-seq peaks

Using HOMER with default settings, peaks for histone marks and Flag tagged Hairy protein were identified using signals from H3 ChIP and input respectively as background.

## Annotation of significantly affected regions

Regions detected by diffReps or peaks called by HOMER were associated with genes by identifying the nearest RefSeq TSS and annotated to a genomic feature such as intergenic, intron, exon etc.

Normalization of ChIP-seq tags for histograms, heatmaps, and scatter plots: We normalized the total number of mapped tags to 10 million for each sample using HOMER so that the read densities were comparable.

## Comparison of ChIP-seq experiments using histograms

ChIP-seq densities of a 4 kb window centered at affected regions detected by diffReps were determined using HOMER. The program normalizes the output histogram such that the resulting units are per bp per region with bin size of 10 bp. Plots were generated using matplotlib (*Hunter, 2007*).

## Comparison of ChIP-seq experiments using heatmaps

Data matrices were generated using HOMER by counting total tags in a 5 kb window around affected regions or peaks and normalizing to 10 million reads with bin size of 25 bp. Data was visualized using Java Tree View (*Saldanha, 2004*).

## Comparison of ChIP-seq experiments using scatter plots

Tag densities were calculated by counting the tags at regions defined by peak coordinates of the first experiment (x axis) and compared to the second experiment (y axis). Data was log2 transformed and plotted using matplotlib. Pearson's Correlation Coefficients were calculated to determine the extent of similarity between samples.

## Analysis of co-occurrence of differentially changed regions

mergePeaks program of HOMER was used to find overlapping sites between differentially changed regions of different histone marks upon Hairy induction. These regions were considered as overlapped if changed regions from each experiment share at least 1 bp. Significance of co-occurrence of regions was indicated by natural log p-values using the hypergeometric distribution. Positive values signify divergence.

## Linking affected regions to Hairy binding

Affected regions for chromatin marks were considered as Hairy bound if the nearest gene has at least one Hairy peak. The occupancy of the induced Hairy protein was compared to that of endogenous Hairy binding by conducting ChIP-Seq analysis using the Flag epitope on the inducible protein; a large fraction (40%) of these binding sites were also found in the ChIP–chip study (p = 2.15e-159). Similar Hairy binding motifs were enriched in both data sets, indicating that the induced Hairy protein has similar targeting specificity to the endogenous protein (*Figure 1—figure supplement 1*).

## Machine learning analysis

We used the differential changes of H4Ac, H3K27Ac, H3K4me1, H3K4me3, H3K36me3, H3K9me3 and Pol II in response to Hairy as features for our analysis here. The genomic blocks detected as significantly altered by diffReps are annotated to closest TSS. We considered four features for each of the ChIP-seq data; number of blocks linked to the same gene, range of blocks, fold change of ChIP-seq signal at blocks, and distance of blocks to closest TSS. Four features from ChIP–chip data sets of Hairy (*MacArthur et al., 2009*), CtBP, and Gro (*Nègre et al., 2011*) were used; number of peaks linked to the same gene, width of peaks, peak signal, and distance of peaks to closest TSS. In addition, expression of transcripts in wild-type embryos was included as a feature. In total, these 41 features were collected for 583 genes (241 repressed, 146 activated and 196 unaffected genes; activated and unaffected genes were grouped as nonrepressed genes) in this study. Differentially regulated genes and their fold changes are listed in *Supplementary file 5* and randomly selected unaffected genes are

listed in *Supplementary file 7*. Important features were first identified with four feature selection algorithms (Information Gain, Symmetrical Uncertainty, Chi Square and Relief). Then, to predict genes in the repressed and nonrepressed categories, four classifiers (Support Vector Machine (SVM), $k$-Nearest Neighbors (KNN), Naive Bayes and Random Forests) were employed. To perform this analysis, we wrote Python and Java codes to partition our dataset into 10 parts to perform feature selection and 10-fold cross validation classification utilizing the Weka machine learning software (http://www.cs.waikato.ac.nz/ml/weka/). To increase the robustness of our results we performed 50 iterations of the above procedure and combined the predicted classes for each gene to create a new aggregate predicted class for that gene. Here we took the class that has been predicted more than 50% of the 50 iterations as the predicted class of the gene. We have applied every combination of the four feature selection algorithms and four classification algorithms to the data to obtain the optimal classification methodology for our dataset. The results of our analysis are summarized in the main text.

## Acknowledgements

We thank Carla Margulies for the Rpb3 antisera, Shin-Han Shiu, Monique Floer, and George I Mias for critical reading of the manuscript, John Johnston and the MSU Institute for Cyber-Enabled Research (iCER) for help with the High Performance Computing Center. We thank Samuel Daulton (Colgate University) for his help in the Machine Learning analysis of the data. We also thank the Arnosti laboratory members for useful discussions, and Sandhya Payankaulam for assistance with establishing library preparation protocols.

## Additional information

### Funding

| Funder | Grant reference | Author |
| --- | --- | --- |
| National Institutes of Health (NIH) | GM056976 | David N Arnosti |

The funder had no role in study design, data collection and interpretation, or the decision to submit the work for publication.

### Author contributions

KK, Conception and design, Acquisition of data, Analysis and interpretation of data, Drafting or revising the article; AA, Performed machine learning analysis; LML, Designed and acquired microarray data, Analysis and interpretation of data; DNA, Conception and design, Analysis and interpretation of data, Drafting or revising the article

## Additional files

### Supplementary files

• Supplementary file 1. GO analysis of down-regulated genes.

• Supplementary file 2. GO analysis of up-regulated genes.

• Supplementary file 3. Comparison of ChIP-seq signal around differentially changed histone marks using Kolmogorov Smirnov test.

• Supplementary file 4. Feature ranking in predicting gene expression.

• Supplementary file 5. Diffentially regulated genes identified by microarray analysis.

• Supplementary file 6. Summary of sequencing reads.

• Supplementary file 7. Randomly selected unaffected genes for machine learning analysis.

## Major datasets

The following dataset was generated:

| Author(s) | Year | Dataset title | Dataset ID and/or URL | Database, license, and accessibility information |
|---|---|---|---|---|
| Kok K, Ay A, Li L, Arnosti DN | 2015 | Data from: Genome-wide errant targeting by Hairy | http://dx.doi.org/ 10.5061/dryad.cv323 | Available at Dryad Digital Repository under a CC0 Public Domain Dedication. |

The following previously published datasets were used:

| Author(s) | Year | Dataset title | Dataset ID and/or URL | Database, license, and accessibility information |
|---|---|---|---|---|
| MacArthur S, Li X-Y, Li J, Brown JB, Chu HC, Zeng L, Grondona BP, Hechmer A, Simirenko L, Keränen SVE et al, | 2009 | Developmental roles of 21 Drosophila transcription factors are determined by quantitative differences in binding to an overlapping set of thousands of genomic regions | http://bdtnp.lbl.gov/Fly-Net/SearchChipper? first=45 | doi:10.1186/gb-2009-10-7-r80. |
| Nègre N, Brown CD, Ma L, Bristow CA, Miller SW, Wagner U, Kheradpour P, Eaton ML, Loriaux P, Sealfon R et al, | 2011 | A cis-regulatory map of the Drosophila genome | http://data.modencode. org/?Organism=D.% 20melanogaster | doi:10.1038/nature09990. |

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
