## [Decision Letter]

[Editors’ note: this article was originally rejected after discussions between the reviewers, but the article was accepted after an appeal against the decision and further revisions.]

Thank you for choosing to send your work entitled “Genome-wide futile cycling by Hairy repressor suggests mechanism for evolution of gene regulatory networks” for consideration at *eLife*. Your full submission has been evaluated by Diethard Tautz (Senior editor), a Reviewing editor, and three peer reviewers, one of whom, Michael Eisen, has agreed to share his identity. The decision was reached after discussions between the reviewers. Based on our discussions and the individual reviews below, we regret to inform you that your work will not be considered further for publication in *eLife*.

This manuscript provides an extensive ChIP-seq data set to study how Hairy regulates gene expression. Using a Hairy over-expression system, authors find that Hairy effects gene expression in a context dependent manner on a subset of target genes. However, on the majority of the bound sites Hairy is able to induce chromatin changes without altering gene expression. Based on these and other observations, authors also speculate about how novel transcription circuits could evolve.

The overall consensus among the reviewers was that although this manuscript does contain some novel observations, it does not provide sufficient mechanistic insights that would explain why only a fraction of Hairy bound sites show a functional correlation and how Hairy induces widespread chromatin changes without changes in gene expression. The fact that the major conclusions are based purely on an over-expression system also poses a major problem. For example, it would be advisable that the authors use other experimental approaches such as depletion experiments to further potentiate their results. The reviewers were also not convinced by evolutionary claims made in the manuscript. The manuscript also lacked a number of controls that made it difficult to assess the quality and strength of the observations.

Reviewer #1:

This is the latest in a series of papers from the Arnosti lab looking at the mechanism of action of the *D. melanogaster* early embryonic transcriptional repressor Hairy. This paper uses a lot of ChIP-seq data to make a fairly simple point: when Hairy binds to sites where it does not appear to be regulating transcription it nonetheless induces changes in chromatin that are similar to the changes it makes when it does have an effect on transcription. This paper makes a valuable contribution. The main point the authors seem to be trying to make – that Hairy affects chromatin even when it's not affecting expression – is lost in the clutter. I think this could be an *eLife* paper, but not in its current form.

What they did:

Used a previously generated transgenic line that expresses Hairy ubiquitously in the embryo upon heat shock, and characterized changes in expression in the Hairy-ub line compared to wt. Compared gene expression in induced Hairy-ub line to control (unclear what control was). Carried out ChIP-seq with H3, H4Ac, H3K27Ac, H3K4me1 and H3K4me3 on wt and Hairy-ub embryos. Carried out similar experiments using a mutant form of Hairy that does not interact with CtBP.

What they claim:

Hairy binding is associated with transcriptional repression, histone deacetylation and changes in histone methylation.

The effects of Hairy are mediated to some extent by its interaction with CtBP.

There are significant alterations upon induction of Hairy in regions where the binding of Hairy does not seem to effect transcription.

Genes could come under Hairy control by the acquisition of activator binding to regions already poised for Hairy repression (although they offer no evidence for this claim).

Concerns:

1) Maybe I'm missing something (though I've looked extensively), but as far as I can tell, nowhere in the paper does it say what the control for gene expression and ChIP experiments was. There are at least three possibilities I can think of, and all of them differ in important ways. Wild type embryos grown under normal conditions? Wild type embryos heat shocked in parallel with the transgenic ones? Transgenic embryos that weren't heat shocked?

[29] use heatshocked wild type embryos, but this also wasn't discussed much in the paper. This paper needs a clear statement of what the control samples were and some indication of why that particular control was chosen.

2) A related issue is that the paper doesn't say in the Methods how long the heatshock was, other than that it was short. It's known that heatshock accelerates development. Since gene expression and chromatin marks at this time are known to be highly dynamic, any chance in the stages present with and without heatshock could produce anomalous results. Same could be true for changes that are the result of the induction of h. It would be nice to see some data on the stages represented in the two samples.

3) ChIP-seq data was normalized to 10,000,000 mapped tags for each sample. However it's easy to imagine that altering the levels of an important transcriptional repressor across the whole embryo could have a global impact on the levels of some of these marks. This really calls for some kind of reference standard that would allow a quantitative comparison of overall levels not just their relative distributions, which is what is being done here.

4) There are many claims made associated with Figure 3, and elsewhere, that Hairy binding is associated with the increase/decrease of some measurement. For example, the claim that there is a significant association of Hairy binding with H4 deacetylation as compared with H4 acetylation as compared with the genome-wide changes in H4 acetylation state. This is based on the observation that, genome-wide, there are a roughly equal number of gains vs. losses of H4 acetylation, but there is a strong bias towards deacetylation in Hairy bound regions. However, it matters a lot what the distribution of H4 acetylation is in Hairy bound regions in controls. If Hairy bound regions tend to have relatively high levels of H4 acetylation, then, even if there is no association between Hairy binding and the change in H4 acetylation upon induction of Hairy-Ub, you would still expect to see Hairy-bound regions go down more often than up because of where they started. The lack of conditioning on the starting state of Hairy-bound regions (as well as possible differences in their size, etc.) is a serious problem with the analyses presented in the paper.

5) I didn't really get the point of the machine learning section. I'm not saying it should be removed, just that the authors should try to explain why they did it and what they learned from it a bit better.

6) I'm not a fan of the use of the term “futile cycling” here, since there is no direct observation of cycling here. What the authors have shown is that the ectopic induction of Hairy leads to an alteration of chromatin state in places where Hairy binding doesn't seem functional. However it seems quite possible that what is happening at these sites is that Hairy is preventing the creation of a chromatin state that would otherwise occur in its absence, rather than altering a previously existing chromatin state. And thus it's not really cycling.

7) I'm all for speculation in papers, but the evolutionary model could use some fleshing out – especially what makes it interesting and novel. It seems pretty obvious that you could make a new enhancer by adding an activator to a previously existing repressor. This idea that enhancer evolution is easier if you add an activator to a repressor rather than a repressor to an activator has been around for as long as I can remember.

Reviewer #2:

In this manuscript, Kok and colleagues report the genome-wide effects of misexpression of the transcriptional repressor Hairy on five histone modifications, RNAPolII binding, and gene expression in *Drosophila* embryos. One of the main findings is that the effects of Hairy binding are context-dependent: some genes that are bound by Hairy are repressed, and many others are not. This is not surprising, in fact it is true of every transcription factor for which regulated genes and genomic binding sites have been compared (see the papers by Fisher and MacArthur cited in the manuscript, or any other ChIP-chip or ChIP-seq study). What is much more surprising is that the H3K36me and H3K9me3 modifications, which are typically strongly associated with actively expressed and silenced genes respectively, were not correlated with Hairy-induced changes in gene expression in this study. The authors suggest that Hairy works through non-canonical mechanisms to regulate gene expression without changing the characteristic markers of gene expression and repression. If so, this is by far the most novel and significant finding of the study, because it overturns the conventional view of the distribution of these histone modifications. However, the authors quickly leave this aspect of the analysis and return to the other modifications, some of whose changes make more conventional sense – assuming that, unlike H3K36me and H3K9me3, their presence or absence can be interpreted in the usual way.

Another unexpected finding is that most genes that are transcriptionally affected by Hairy overexpression (either directly or indirectly) show no change in PolII binding, either at the promoter or in the gene body. A lack of binding at the promoter could be explained by the idea of “promoter escape” mentioned by the authors (subsection “RNA polymerase II and silencing by Hairy”), but the lack of change of PolII in the gene body cannot. It is possible that these targets are regulated post-transcriptionally (and therefore indirectly), so that transcription is unaffected (which would be consistent the lack of change in PolII, H3K36me and H3K9me3) but RNA levels are reduced (which would be consistent with the microarray data)? Or could the result be due to the significant heterogeneity of the cells in the whole-embryo samples?

Another difficulty with the study is the significant problem of direct vs. indirect regulation by Hairy. The authors note that over 91% of Hairy-bound genes are not repressed, and a significant fraction of repressed genes (31%) are not bound by Hairy. They consider the 167 genes that are both bound and repressed to be directly repressed by Hairy, but this is probably a significant overestimate. Since the vast majority of Hairy binding sites are apparently nonfunctional, is it safe to assume that all of the Hairy sites in down-regulated genes are functional, especially since many other genes are regulated indirectly? How many of these binding sites in the 167 regulated genes are located within active enhancers? How many of the known direct target enhancers of Hairy (not the genes, the enhancers) are bound in this experiment? Compounding the uncertainty, the Hairy binding sites shown in the figures are not from the over-expressed hs-*hairy* used in this study: they are ChIP-chip data from the endogenous protein (30). Only 40% of peaks are shared between the MacArthur dataset and this study (subsection “ChIP-seq”, last paragraph), suggesting that the distribution of over-expressed Hairy is very different from that of endogenous protein, not unexpectedly since hs-*hairy* is expressed in all cells of the embryo, most of which do not normally express Hairy. The measurements in this study, unavoidably, are averages across all cells of the embryo, so it is problematic to correlate a change in gene expression with a lack of change in chromatin modifications: these may be occurring in different cells.

Another concern is that the authors explain an up-regulation of *HLHm7* by H-mut-CtBP (Figure 6) by proposing “antagonism of the endogenous wild-type Hairy by the mutant” (subsection “CtBP supports Hairy repression activity”, second paragraph). If the authors are correct, this totally nullifies the other conclusions drawn from the experiment.

The attempt to predict repressed vs. non-repressed genes using machine learning does not seem to have produced any useful insights, or at least those insights were not made clear in the text (subsection “Predicting a “successful” repression context”) or in Figure 7.

The Discussion section is far too bold in its claims, relative to the results, especially in its evolutionary speculations. To flatly state that “essentially trivial biological interactions have consequences in evolutionary time” is to go far beyond the data. That statement may be true, it probably is true at least some of the time, but it is not a finding or a reasonable interpretation of this study, and there is nothing in this study to support it. I am also confused by the concept of “futile cycling”: I'm not clear on exactly what is cycling or being cycled. If it is chromatin, is there any evidence for higher chromatin turnover dependent on Hairy binding? Another notable statement is that “little distinguishes the types of chromatin effects observed on repressed genes compared to ‘non-functional’ interactions on other loci” (Discussion, sixth paragraph). That is indeed a finding of the study, but at least for me, it is more likely to reflect a problem with the data than to prove that most of what we know about gene expression, silencing, and their associated chromatin modifications is wrong. If the latter is true, the authors have a tremendously important result on their hands, but that will require a much higher level of proof.

Finally, it is quite easy to propose that non-functional Hairy binding sites are “an off-the shelf module that only needs the addition of a few activator binding sites to emerge as a fully fledged *cis*-regulatory element.” However, (1) no experimental support for this idea, direct or indirect, is provided here or cited from other studies (mentioning that changes to enhancers can be evolutionarily important does not count as evidence in favor of this particular proposal), and (2) this evolutionary model does not depend on any of the results shown here, since widespread, apparently non-regulatory binding sites have been commonly observed since the first days of ChIP-chip (e.g., Holloway, Genome Inform 2005; Moses, PLOSCB 2006). This is a group of talented, rigorous, and thoughtful investigators, but in this case their large biochemical and evolutionary claims are not well supported by the data.

Reviewer #3:

Kok and co-workers have carried out an extensive genome-wide analysis of the effect of Hairy on the chromatin landscape in an effort to correlate these effects with transcriptional repression by Hairy. The ChIP-seq analysis in comparison with the transcriptome analysis shows that Hairy binds to many more genes than it represses. Analysis of the chromatin landscape shows that repression is largely associated with histone deacetylation and H3K4me1 demethylation in regions that include the Hairy binding sites and spread from these sites. Perhaps surprisingly, however, genes that are not repressed by Hairy, but that bind Hairy, often show similar changes in the chromatin landscape (what the authors call futile cycles of modification). This shows that deaceylation and H3K4me1 demethylation are not sufficient for repression and even suggests that they could be a consequence as opposed to a cause of repression. Various machine learning protocols are used in an effort to determine what features might be most predictive of repression and some moderate success is achieved. The authors propose that these non-functional Hairy binding sites could facilitate the evolution of new *cis*-regulatory modules through the acquisition of additional transcription factor binding sites. In general, I think this is a well-executed study that brings out some important new ideas. I have a few questions that I would like the authors to address:

1) The authors have carried out the analysis of Hairy targets by comparing Hairy-overexpressing with wild-type embryos. This has both advantages and drawbacks. The advantage, of course, is that it is technically feasible. But the disadvantage is that Hairy is being expressed in cells in which it is never normally present and that this abnormal context could have functional consequences that we can't easily account for. This could potentially account for the lack of correlation between changes in the chromatin landscape and repression. While the ChIP-seq analysis is probably technically out-of-reach in loss-of-function mutants, have the authors considered at least carrying out the RNA-seq analysis in loss-of-function (RNAi knockdown?) embryos. It would be worth knowing how many of the repression targets identified by overexpression are also identified by loss-of-function.

2) I found the lack of correlation between the Pol II ChIP and repression to be puzzling. The authors have not clearly explained how they account for this. Has this lack of correlation between Pol II ChIP and transcription levels been observed in other contexts?

3) I don't fully understand the model in Figure 8. It needs to be explained in the text.

[Editors’ note: what now follows is the decision letter after the authors submitted for further consideration.]

Thank you for resubmitting your work entitled “Genome-wide errant targeting by Hairy” for further consideration at *eLife*. Your revised article has been favorably evaluated by Diethard Tautz (Senior editor), a Reviewing editor (Asifa Akhtar), and two reviewers. The reviewers appreciated that the manuscript has significantly improved upon revision and therefore we would like to go forward with the manuscript. However, before acceptance we would like you to address the remaining concerns (below) adequately in the text so that the readers are aware of the limitations and the conclusions that can be drawn an over-expression system.

Reviewer #1:

I think the authors did an excellent job addressing our concerns. The revised manuscript is far more direct and readable, and I think paints a more accurate picture of what the data do and do not say. There are aspects of it that I would change if it were my manuscript, but I don't think it's the role of reviewers to shape manuscripts entirely to our tastes, so I will refrain from pointing them out.

I think there one overarching question remains, which is that, to me, none of what they report in the manuscript is surprising. It's very much consistent with lots of data from lots of labs over the past decade. However, what the authors show here has not been demonstrated experimentally yet, and, as such, I think it is an *eLife* level contribution. But I can see it either way.

Reviewer #3:

The authors have addressed many of the criticisms in the initial review through additional data analysis and reference to the literature. I have two concerns:

1) The original review asked for loss-of-function data to confirm the results of the overexpression studies. In response, the authors cited previous literature showing that hairy loss of function results in derepression of many of the genes that are repressed upon hairy overexpression. However, this does not address the more important point that pertains to the genes that exhibit changes in the chromatin marks and yet show no repression upon hairy overexpression. This is the class of genes upon which the claim to novelty is based and therefore, this is the class of genes whose behavior needs to be verified in loss-of-function studies.

2) I find the machine learning section of the proposal a little puzzling. On the one hand the authors claim that repressed and non-repressed genes often show similar changes in histone marks upon Hairy overexpression. However, the machine learning section seems to suggest that the changes in chromatin marks are not really the same for repressed and non-repressed genes. How do the authors reconcile these apparently disparate claims?

---

## [Author Response]

[Editors’ note: the author responses to the first round of peer review follow.]

Thank you very much for guiding the editorial process of our recent manuscript. We read the reviews carefully, and discussed the main points with two individuals who agreed to be identified. The comments were quite helpful in focusing and clarifying the work; additional analysis on our part was able to address each of the points raised in the review process. […] A major new point that came from responding to questions in this review process was the consideration of how the “errant targeting” and biochemical activity of Hairy, as well as potentially many other factors, calls into question our assumptions in calling enhancers from genomics datasets. This, in addition to consideration of evolution of CREs, will be of great interest to *eLife* readers.

We address the comments from Reviewers 1, 2 and 3 below.

Reviewer 1 found the presentation complicated and confusing; in part because we presented two parallel narratives. One dealt with the implications of genome-wide chromatin modifications by Hairy, while a separate one focused on the biochemistry of Hairy and role of the CtBP corepressor. We agree with the reviewers that the main focus was on the former story, and accordingly we changed the structure of the paper to reflect this focus, removing data about analysis of the CtBP binding mutant which overly complicated the presentation. To facilitate presentation of our extensive chromatin modification data sets, we now present individual points in separate figures (Figures 2 and 3, Figures 4 and 5; previously combined in Figures 2 and 3 respectively). We also extensively revised the model in Figure 8 (now Figure 9) to make it easier to understand.

Reviewer 1 raised a concern about the controls used for inducible gene expression in the ChIP experiments. In the old version of the manuscript, we had cited the controls used for embryo collection and heat-shock treatment (29). To clarify this point, in the revised version, we explain in the Results section: “We treated the control embryos identically to embryos carrying the inducible Hairy transgene to control for possible nonspecific effects of heat shock on gene expression and chromatin marks. In this system, heat shock alone has no effect on the expression patterns of the pair rule and other genes analyzed, and the chromatin marks in heat shocked control embryos were indistinguishable from chromatin patterns previously reported for untreated embryos ([29]; The [34]).” Examples showing how the embryonic chromatin from our samples (heat shock treated) closely resembles that of modENCODE embryos (H3K27Ac, H3K4me1 and me3), and even to some extent from S2 cells (H4Ac) are shown in Figure 10

Author response image 1.Comparison of modENCODE data with our data from control embryos not expressing Hairy that were heat shock treated. The high level of similarity for the genome tracks of H4Ac, H3K27Ac, H3K4me1 and H3K4me3 at two loci were representative of the overall similarity across many genomic loci.**DOI:**
http://dx.doi.org/10.7554/eLife.06394.028

We plotted genomic data using reads normalized to the sequencing depth (normalized to 10 million). Reviewer 1 asked whether induction of Hairy might cause a global impact on histone marks, which would be missed by after normalization. We considered this possibility; in fact, we find that whether or not Hairy is induced, the genome features for multiple chromatin marks are virtually identical, except in very discrete regions where there are significant changes, as shown in Figure 2—figure supplement 1), which shows the high similarity in histone marks between control and Hairy induced embryos on a specific locus (A) and globally (B and C). We now emphasize this point in the Results for clarification.

Reviewer 1 noted that the tendency for Hairy-bound regions to show loss rather than gain of acetylation/methylation marks may be influenced by the starting level of these chromatin marks. If Hairy is overrepresented in regions highly enriched for these marks, then there might be a bias. However, this is not the case. For H4Ac, the majority of Hairy reduced regions were not in the ∼2700 regions called as peaks by HOMER, suggesting that Hairy is not preferentially acting in “hot spots” for this modification (Figure 2—figure supplement 1). Furthermore, for differentially reduced levels of H4Ac, the average level of this modification in regions not bound by Hairy was actually higher than the average level for regions on which Hairy specifically acts (Figure 3—figure supplement 1, blue plots). Similar relationships hold for H3K27Ac and H3K4me. These results indicate that the tendency for Hairy-bound regions to show a loss in histone marks is not simply due to a biased high starting state.

Reviewer 1 asked that we better explain the purpose of carrying out machine learning on our chromatin data, and what we learned from it. The variety of chromatin alterations induced by Hairy did not contain a simple pattern, such as magnitude of change or width of affected area, predictive of transcriptional repression. Clearly, the ability to repress or not might hinge on where activators are bound; we didn’t measure the location of all activators, as this is not currently available knowledge. However, using machine learning, we did test the hypothesis that certain combinations of factors relating to position, size of chromatin block, magnitude of changes, occupancy of Pol II and other factors would tend to differentiate “real” regulatory events from “simulations”. The “Random Forests” method, shown in Figure 8, was able to perform substantially better than random guessing for calling both “repressed” and “nonrepressed” genes (e.g. for repressed, 75% success, vs. 40% for random guesses), indicating that some special combination of certain features – as discussed in the Results – do provide a partial indication of where chromatin alterations are more likely to be associated with changes in gene expression. The overall success rate is still well below 100%, however. We changed the figure legend in Figure 8 to show on the heat map which level exceeds the performance of random guessing. Other machine learning approaches were able to make useful predictions, but were less successful; such differences in performance are routinely found for different sorts of datasets, and usually machine learning studies do not dwell on the weaker algorithms. We have revised this section to emphasize that there are at least two kinds of information relevant for interpreting “real” from “background” chromatin signatures, one of which is directly accessible from this sort of study. The importance of this point is that many genome-wide analyses simply assume that a particular set of modifications proves that there is a regulatory element at a locus – this is an erroneous and misleading way to approach genome-wide information. We address this this point with revised text in Results and Discussion.

Reviewer 1 was not in favor of using the term “futile cycling” for the biochemical changes observed on non-functional Hairy-bound sites. Although we noted in the manuscript that the changes are indeed impermanent “Hairy repression is readily reversible, with chromatin reverting to its previous state minutes after depletion of the overexpressed repressor” we agree that this term may be misleading. Therefore we use “genome-wide errant targeting by Hairy” as now employed in the revised title and text.

Reviewer 1 urged us to flesh out the evolutionary model, noting that the idea of evolutionary appearance of binding motifs to make new *cis*-regulatory elements is not new. We agree that this part of the argument was weak; removing the CtBP – related data allows us to focus directly on this point. We now present in the Discussion (and provide relevant background information in the Introduction) two major points; first, previous studies had suggested that Hairy as a repressor only functions in the context of an intact and active enhancer. Our work demonstrates that this protein is able to mediate its biochemical activity in most bound regions, indicating that there is little context necessary for the protein to go to work. Second, much molecular biology research has emphasized the high degree of cooperativity necessary for metazoan transcription factors to work well. Enhanceosomes, patterning elements and other enhancers give aberrant readouts if correct stoichiometries and spacings are not respected. These findings suggest that random individual sites are less likely to generate a suitable transcriptional readout. At least for the repressor side of the equation, we see here that the demands for generating biochemical activity are lower than anticipated – our work challenges our colleagues in the field to determine the generality of previous assumptions, and measure the independent potential for many activators binding to “off target” sites.

Reviewer 2 notes that one of the main findings presented is that Hairy interacts with many “off target” sites, with relatively few linked to transcriptional function. The reviewer notes that such observations have been made previously, and this finding is hardly surprising. We completely agree, and in fact cited previous studies discussed this point (30; 13; 11). As emphasized in our manuscript, it is not the presence of off-target binding that is novel, but the realization that these events are linked with extensive biochemical activity, in contrast to previous assumptions, and the biological implications of this finding – namely, 1) enhancers may have a lower threshold for formation that we might have expected, and 2) genome-wide studies that blithely assume a change in chromatin mark “proves” you have an enhancer at some locus are making an unwarranted assumption.

Reviewer 2 was surprised that genes silenced by Hairy lacked certain canonical chromatin changes previously suggested to be linked to repression, namely, loss of H3K36me3 and gain of H3K9me3. The reviewer notes that this would overturn the conventional view of how these histone modifications work, and asked why we didn’t comment on this point. Indeed, our initial draft did not comment on this apparent discrepancy, an omission that we have now remediated in the revised Results. Although transcriptional review articles have pointed out general correlations between individual marks and gene expression, the actual literature clearly shows that such correlations are not strong, and there is considerable heterogeneity in how they are involved in gene expression. We now cite some of these studies; for example, upregulation of KDM4A target genes in *Drosophila* occurs without increases in H3K36me3 (10). Similar studies with Spt6 in *Drosophila* further indicate that H3K36me3 levels do not correlate with *Hsp70* gene expression (2)**.** In fact, H3K36me3 may in some contexts contribute to gene silencing due to its presence in heterochromatic domains (8) and in other cases, removal of H3K36me3 is required to promote transcriptional elongation (25). Similarly, repressive marks such as H3K9me3 and H3K27me3 are not always simply coupled to repression. For example, only a modest correlation between H3K9me3 and H3K27me3 levels and gene silencing was observed (5; 54). In differentiation of T and B cells, a small fraction of repressed genes ever acquired H3K27me3 (54; 33). Interestingly, H3K9me3 was found to be enriched in many active promoters and associated with transcriptional elongation (48; 51). In summary, the simple correlations previously noted mask much more complex pictures that are revealed by analysis of genome-wide datasets, such as ours. We think that these complexities are themselves intriguing, and now have expanded our Discussion to provide more context.

Reviewer 2 was intrigued by the lack of change in Pol II occupancy on the bodies of many repressed genes, and offered suggestions on why this may be so, including a possible post-transcriptional regulation that would identify genes as “repressed” when in fact it was enhanced mRNA turnover at work. The short time span from Hairy induction to collection of mRNA makes it less likely that posttranscriptional effects mediated through Hairy targets would account for this effect, however. We favor the second hypothesis offered by the reviewer, namely, that we measure transcription in embryos with heterogeneous nuclei. Genes that feature poised polymerase at the promoter in many or most nuclei, but are only expressed in a few nuclei will have strong Pol II promoter signals but weak signals at the body of the gene, as observed on the *gogo*, *pros*, and *tup* genes shown in Figure 7. Therefore, in this view, the lack of change in Pol II levels on the gene body reflects the inherently low signal, rather than a biochemical mechanism. This explanation accounts for a considerable number of affected genes. In addition, Hairy is associated with many repressed genes with higher gene body signals where Pol II levels do decrease (Figure 7). However, Hairy is also associated with many repressed genes where Pol II levels can be measured on the gene body, and these levels are unchanged even as mRNA levels drop. These different observations suggest that depending on context, repression may differentially impact Pol II properties. Such mechanistic heterogeneity has been previously observed; in the revised Results, we cite several examples where reduction in gene expression does not result in loss of Pol II from the gene body (53; 1; 2; 29). We now note that Hairy may interfere with gene expression at different steps of the transcription cycle as suggested for repression by GR, indicating gene specific repression mechanisms (18).

Reviewer 2 raised a point about whether target genes that were repressed and bound by Hairy in vivo were in fact direct functional targets of the protein, since some repressed genes were not bound by Hairy, and many genes bound by Hairy are not repressed (consistent with our findings that Hairy has many nonfunctional sites). 1. The reviewer asked whether it is safe to assume that all of the Hairy sites in downregulated genes are functional. Indeed, to assume that every single bound site must be functional is unlikely, however, importantly, our study does not rest on that assumption, only that the large majority of target genes are bona fide Hairy targets based on the criteria below. 2. The reviewer asked how many of the Hairy binding sites on the 167 genes scored as direct targets overlap with active enhancers. High-quality data about validated enhancers remains to be determined for most of these genes, but if we use ChIP occupancy of transcription factors as a proxy for “active enhancer” (which is an admittedly soft proposition), and map Hairy sites within these elements, we find that 163 of these 167 genes has Hairy associated with a putative regulatory module (data from [30]; Kok unpublished observations).

We emphasize that the current state of knowledge in this field is still too fragmentary to decisively answer this question. 3. The reviewer asked how many known Hairy target enhancers are bound in this data set. Most functional studies addressing enhancer interactions with Hairy have been performed with synthetic *cis* elements, however, Hairy direct regulation of *ftz* 5’ and 3’ enhancers has been reported, and we find that these enhancers are indeed bound by Hairy (29). We also identify the *Sxl* promoter region as a direct target bound by Hairy; this gene has previously been characterized as a *deadpan* target, which has a similar bHLH DNA binding domain to Hairy, and Hairy is shown to regulate this gene (Dawson et al. 1995). There are no other reports of Hairy interacting with natural enhancers, where the role of Hairy has been described at a molecular level; our study provides the first elucidation of other possible targets. 4. The reviewer asked whether the use of ChIP data for non-overexpressed Hairy was suitable, as only 43% of our Flag epitope signals for the overexpressed Hairy protein overlapped with the ChIP-chip data. We selected the ChIP-chip data for our analysis because the Flag epitope gave low signals overall, although high-confidence functional targets such as *ftz, Impl2, odd, h, 18w, wg, tup, pros, nht,* and *en* were found. Practitioners of the trade know that there are often discrepancies in the exact sets of genes captured by ChIP studies, but the bulk of evidence from this study strongly supports the set of genes as a whole being likely Hairy targets; first, the overlap between repressed genes and genes bound in vivo is highly significant (p=3.8e-95), second, down-regulated genes are significantly enriched in classes consistent with known Hairy function ([Supplementary-material SD1-data]), including transcriptional regulation, cell fate commitment and neurogenesis (p<3.7e-18), third, many of the predicted target genes including *odd*, *comm*, *comm2*, *edl*, *en*, *Impl2, prd*, and *18w* have striped expression patterns complementary to Hairy’s, and fourth, several of these have been shown to be derepressed in *h* mutant embryos (22; 6). In sum, we have good evidence that down-regulated genes are likely to be direct Hairy targets, and based our conclusions not on the activity of any individual gene but on the groups as a whole. Uncertainty about small numbers of genes would not significantly change the conclusions. We revised the Results to include information about the correspondence of the genetically confirmed Hairy targets and our molecular results.

Reviewer 2 raised a question about derepression of *HLHm7* upon induction of the CtBP mutant version of Hairy. Since we removed this section from the manuscript as described above to streamline the presentation, it is not a part of the Discussion anymore.

As did Reviewer 1, Reviewer 2 asked about the purpose of the machine learning study, which we address above.

Reviewer 2 raised questions about how we handled the discussion of evolutionary implications of this study, noting that we do not show examples of how these “off target” biochemical interactions may facilitate the generation of novel *cis*-regulatory elements. We agree that, buoyed by the novelty of the findings, our description did indeed veer heavily into a speculative mode. We have made significant changes to the Abstract and Discussion to tone things down a bit and point out what we know and what the field should be investigating as possible leads. At this point, we do not have in hand an example where a Hairy bound “off target” site was subsumed into a novel *cis*-regulatory element. Studies that identify new enhancers have been carried out successfully in flies only to extent of showing how new genes (in male reproduction, for example) can pick up *cis*-regulatory elements from a variety of exapted ancestral or novel DNA elements. Alternatively, very beautiful individual cases have been described showing how subfunctionalization or neofunctionalization of *cis*-regulatory regions plays a role in evolution of gene expression. These studies fail to provide much information about which factors are present on individual elements, or how their activities may be relevant to generating the new element. Our study highlights the need to consider the activity of “off target” sites in generating novel elements, particularly because for Hairy at least (and likely other factors that employ the same cellular machinery) they are “shovel ready” and not constrained by complex *cis*-regulatory grammar. The new version of this manuscript provides a more balanced discussion of what we know and what we don’t know, and now develops a second theme that was stimulated by considering your comments. That is, genomic consideration of chromatin marking must not equate changes in some active marks with enhancers – there appears to be a significant possibility for false positives if one relies on the simple correlations that are widely employed.

The reviewer was confused by the term “futile cycling”. We recognize that this was not a helpful designation, and have altered this, as described above.

“H” was used in Figure 1 to denote Hairy protein; the reviewer suggested we use hs-*hairy* to distinguish this from Hairless, which is another gene. We have modified the figure accordingly.

Reviewer 3 noted that induction of Hairy in cells in which it is never normally present could induce artifacts, accounting for the low correlation between changes in the chromatin landscape and repression. We agree that it is possible that some loci identified are not real Hairy targets, but only when the protein is overexpressed. However, for hundreds of the loci considered where chromatin changes are taking place, ChIP data from both Biggin as well as Kevin White confirm that Hairy can be found binding to these genomic regions. The early point in development selected for our study means that there are fewer cell-specific modifications, Polycomb repressed regions, and other markers of differentiated or senescent cells; thus the heterogeneity question is of lesser concern. We now do make a remark about this consideration in the Discussion, however.

Reviewer 3 recognized that the overexpression system allows us to carry out manipulations that would be otherwise impossible, but asked if we can also use knockdown approaches. The inducible system allows us to capture direct effects with good temporal resolution, which would not be possible in loss of function assays such as RNAi knockdown. In response to this suggestion, in the current version we note *h* loss of function mutants cause derepression of genes such as *ftz*, *en, edl, Impl2,* and *prd* as shown in previous studies, and these same genes are down-regulated upon Hairy induction in our hands (22; 6).

Reviewer 3 points to the lack of correlation between the Pol II ChIP and repression and asks if it has been observed in other contexts. We addressed this question in our response to Reviewer 2.

Reviewer 3 asked that Figure 8 (now Figure 9) be better explained in the text. We made a new figure for the model and clarified the Discussion.

[Editors’ note: the author responses to the re-review follow.]

We appreciate the reviewers’ careful reading and helpful suggestions. We address the comments from Reviewers 1 and 3 below.

Reviewer 1 noted the revised manuscript is more direct and readable; we thank the reviewers for their feedback that aided us in improving this paper. The reviewer noted that data from many groups are consistent with our findings, but this study provides for the first time direct experimental evidence for “errant targeting” that results in direct biochemical modification of chromatin regions, not necessarily linked to changes in expression. We agree with the reviewer that our findings indeed have a bearing on numerous ChIP-seq studies.

In the first round of review, Reviewer 3 noted that overexpression used here means Hairy will be expressed in (interstripe) cells not normally containing the protein. There, the Reviewer asked if knockdown studies might indicate “how many of the repression targets identified by overexpression are also identified by loss of function”. We interpreted this point as asking whether the repressed genes found in our study were corroborated by other methods, and cited previous literature supporting the identification of a number of Hairy targets. In response, in the second round the reviewer noted that “this does not address the more important point that pertains to the genes that exhibit changes in the chromatin marks and yet show no repression upon hairy overexpression”. Based on this comment, I think that we misinterpreted the thrust of the question in the first place, so here we address the question of the unaffected genes experiencing chromatin modification by Hairy. We have previously discussed data indicating that many of such elements are known to be occupied by endogenous Hairy. The reviewer asked that the behavior of this class be verified in loss-of-function studies. As previously addressed, the ChIP-seq/chromatin change studies require the quick kinetics that overexpression but not knockdown allow, but what might depletion of Hairy reveal? We thought about the possible mechanistic models possible in this case. 1) A gene normally bound by Hairy in the nuclei within the blastoderm stripes whose global expression is unaffected despite induction of Hairy may have separate enhancers that are far from Hairy sites. Even when Hairy binds in the interstripe nuclei, the enhancers would still escape regulation. 2) Alternatively, the gene may be silent, in which case Hairy binding to the gene in interstripe nuclei would not alter its silence. 3) A third possibility is that within the normal Hairy stripe regions, there are special activators that are only present in these stripes, but the endogenous Hairy is at high enough levels so that expressing more Hairy has no effect on transcription – the observed chromatin changes would come only from interstripe nuclei. Only in this last case would we mistakenly “call” a gene as “not regulated transcriptionally” but still having chromatin modifications. We think that there are few genes that fit this third category; known activators at this stage tend to be more broadly expressed, and loss of function *hairy* mutants show widened expression of target genes (stripes expand), consistent with broadly expressed activators. These scenarios are shown in Figure 10. We know that hundreds of genes we consider are broadly expressed, thus the conclusions drawn are based on this more general picture. To be frank, we had not considered all of the scenarios before this point was raised, and it is a useful one. We elaborated this point in the Discussion so that the readers are aware of these issues.

Author response image 2.A) Three scenarios explaining why genes might be unresponsive to Hairy induction in interstripes and stripes even though chromatin changes were detected. 1. Enhancers are far from Hairy sites. 2. Gene is generally inactive. 3. Stripe-specific activators are present within Hairy stripe region, and are partially or completely repressed by the endogenous Hairy, so that ectopic Hairy has no effect on transcription. Promoters are shown in Hairy stripe and interstripe areas, before and after Hairy induction. B) Diagrams show expression pattern of Hairy (red) and Type 3 unresponsive genes (blue) to Hairy overexpression in wild-type (wt) embryos, loss-of-function and overexpression assays.**DOI:**
http://dx.doi.org/10.7554/eLife.06394.029

Reviewer 3 pointed out the apparent contradiction between our assertions that Hairy action on repressed and non-repressed loci was similar, yet machine learning methods can differentiate promoters that are transcriptionally repressed from others.

The two settings for these conclusions are somewhat different. For machine learning, we classified the gene sets as repressed and nonrepressed (activated and unaffected), whether or not they are bound by Hairy or affected at the chromatin level in a particular manner. The machine learning algorithms predict gene expression with up to 75% accuracy, not perfect but much better than random guessing. Separately, in the Discussion, we pointed out the similarity of chromatin changes on targets bound by Hairy, whether or not they were transcriptionally regulated. We address these differences in the revised manuscript, where we clarify these points in the text and Figure 8 (subsection “Predicting a “successful” repression context” and modified labeling in Figure 8).